# BioKGBench: A Knowledge Graph Checking Benchmark of AI Agent for Biomedical Science

## Abstract

Pursuing artificial intelligence for biomedical science, a.k.a. AI Scientist, draws increasing attention, where one common approach is to build a copilot agent driven by Large Language Models (LLMs). However, to evaluate such systems, researchers typically rely on direct Question-Answering (QA) to the LLM itself or through biomedical experiments. How to benchmark biomedical agents precisely from an AI Scientist perspective remains largely unexplored. To this end, we draw inspiration from scientists' crucial ability to understand the literature and introduce BioKGBench. In contrast to traditional evaluation benchmarks that focus solely on factual QA, where the LLMs are known to have hallucination issues, we first disentangle "Understanding Literature" into two atomic abilities: i) "Understanding" the unstructured text from research papers by performing scientific claim verification, and ii) interacting with structured Knowledge-Graphs for Question-Answering (KGQA) as a form of "Literature" grounding. We then formulate a novel agent task, dubbed KGCheck, using KGQA and domain-based Retrieval-Augmented Generation (RAG) to identify factual errors in existing large-scale knowledge graphs. We collect over two thousand data points for the two atomic tasks and 225 high-quality annotated samples for the agent task. Surprisingly, we find that state-of-the-art general and biomedical agents have either failed or performed inferiorly on our benchmark. We then introduce a simple yet effective baseline, dubbed BKGAgent. On the widely used popular knowledge graph, we discover over 90 factual errors which provide scenarios for agents to make discoveries and demonstrate the effectiveness of our approach.

## 1 Introduction

Large Language Models (LLMs) are so powerful that they facilitate nearly every aspect of daily life and work right now, even research (Zhao et al., 2023; Baek et al., 2024; He et al., 2023; Zhou et al., 2023). Observing their marvelous successes in text generation (Yu et al., 2022; Celikyilmaz et al., 2020), text summarization (El-Kassas et al., 2021; Gambhir & Gupta, 2017), and other tasks (Jin et al., 2024a; Tang et al., 2023a), along with their consistent failures such as hallucinations (Ji et al., 2023; Yao et al., 2023), one can conclude that LLMs are powerful in certain tasks involving large-scale unstructured data like daily text or images, but relatively powerless when dealing with data-hungry scenarios. As such, researchers then construct AI agents (Wu et al., 2023b; Tian et al., 2023) assisting LLMs with external tools to extend the capabilities of LLMs. These attempts are fruitful in many fields, including autonomous computers (Steiner, 2008), shopping web-agent (Lee & Liu, 2004), code development (Dalle & David, 2004), society simulation (Drogoul & Ferber, 2018; Lan et al., 2023), etc. A natural subsequent attempt is to develop AI agents to simulate scientists, aiding or even taking over the process of scientific discovery (Baek et al., 2024).

As in Figure 1, existing attempts can be grouped into two categories: i) to build an AI agent for a specific task, such as Question Answering (QA) in a specific domain (Zhang et al., 2018); ii) to encompass multiple AI agents to formulate a multi-agent system as the copilot of scientists, automating certain scientific activities, such as experiment result analysis (Bi et al., 2023; Wang et al., 2023b).

Literature review is the most critical ability that a scientist should possess (Snyder, 2019; Thomas et al., 2020). It does not only involve reading and memorizing, but also requires scientists to

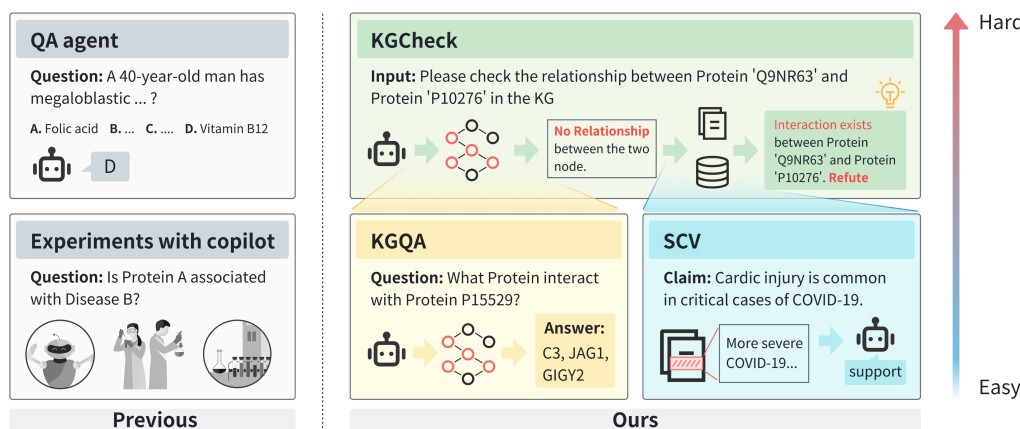

Figure 1: **(Left)** Previous benchmarks for domain-specific AI Agents either focus on the low-level tasks like question answering or are embedded in a complicated pipeline as a scientist copilot. **(Right)** We close the gap by constructing a knowledge graph checking task that consists of two atomic sub-tasks: Knowledge Graph Question Answering (KGQA) and Scientific Claim Verification (SCV), to provide a better evaluation of AI Agents in biomedical science domain.

understand and critically analyze. Researchers and scientists widely spend a significant amount of time in reading recent works. To save human efforts in scientific discovery, it is necessary for AI scientists to be able to accurately understand and analyze the existing research. Many researchers have dedicated to literature understanding in AI agents (Cai et al., 2024; Li et al., 2024), while a systematic evaluation system is missing and even underexplored. The current finest evaluation system (Cai et al., 2024) consists of multiple-choice questions extracted from literature, which cannot fully reveal the underlying reasoning regime of an agent's success or failure, leaving no clue for future advancement nor indicating whether the agent understands the reasoning rationale or merely memorizes data patterns.

On the other hand, another crucial research direction is to help AI agents capture the underlying logic of literature through domain-specific Knowledge Graphs (KGs) (Abu-Salih, 2021; Kejriwal, 2019). KGs store massive knowledge triples in a graph-structured format (Hogan et al., 2021; Alqaaidi & Kochut, 2024), complementing LLMs with external knowledge while providing frameworks for interpretation and reasoning (Meyer et al., 2023). However, manually constructing such KGs is both intellectually and physically intensive. These domain-specific KGs require annotators with profound domain-specific knowledge, leading to high costs to create or maintain the knowledge graphs. As such, we observe that the existing and well-known biomedical KGs (Santos et al., 2022; Chandak et al., 2023) are not fully reliable due to outdated information. We attribute such discrepancy to the static nature of KGs, which lack mechanisms for dynamic updates to align with the evolution of external knowledge sources.

In this paper, we propose a novel agent evaluation benchmark BioKGBench to address both challenges simultaneously. As in Figure 1 (right), the ultimate goal of our benchmark is to verify the correctness of nodes and triples in the knowledge graph based on various information, including papers and well-maintained databases. We dub this task Knowledge Graph Checking (KGCheck). Agents need to first query the information recorded on the KGs as directed, then cross-reference this information with external literature or databases to combat hallucinations. This task evaluates the agents' capacities to both process and understand structured data (like KGs) and unstructured data (like literature). It is worth mentioning that the process of verifying knowledge within KGs closely mirrors the methodology of human scientific research, including database queries and extensive literature reviews. This similarity not only underscores the task's relevance to real-world scientific inquiry but also provides intriguing insights. Furthermore, we decompose this task into two more atomic subtasks: Knowledge Graph Question Answering (KGQA) and Scientific Claim Verification (SCV), enable a more detailed evaluation of the agents' capabilities in processing and understanding of structured and unstructured data, respectively.

We extensively analyze existing AI agents on our benchmark and find that none of the existing agents can accomplish our tasks without moderate adaptation. Therefore, we introduce our agent BKGAgent, the first agent framework to interact with external knowledge graphs as well as research papers. Experiments demonstrate fascinating results that our agent is capable of discovering real conflicts in the existing large-scale datasets. Within 225 professional-annotated data in the Clinical Knowledge Graph (CKG) (Santos et al., 2022), our agent BKGAgent successfully identified some conflicting or missing pairs. This evidence further supports the academic value of our agent by providing researchers with a tool to update their own knowledge bases, offering substantial potential in both academic and commercial markets.

## 2 RELATED WORK

**Science Agent.** The swift progression of large language models (LLMs) has catalyzed the widespread deployment of intelligent agents across diverse fields, notably within the science domain. Notable examples include ChemCrow (Bran et al., 2023) and Coscientist (Boiko et al., 2023b) in the field of chemistry, DoInstruct (Bi et al., 2023) in ocean science, and GeneGPT (Jin et al., 2024b), Almanac (Zakka et al., 2024), MedAgents (Tang et al., 2023b) in the biomedical domain, etc. Among them, biomedical agents, in particular, have garnered significant attention due to their critical importance. Biomedical agents (Gao et al., 2024) impact areas ranging from hybrid cell simulation (Xiao et al., 2024), the design of cellular circuits (Chandrasekaran et al., 2024) to the development of new therapies (Zhenzhu et al., 2024) and so on. We posit that biomedical agents will emerge as a focal point of research. However, the current benchmark in this field remains inadequate. For instance, MedAgents is evaluated in MedQA (Zhang et al., 2018), MedMCQA (Pal et al., 2022), PubMedQA (Jin et al., 2019), relying heavily on inherent knowledge of LLMs, which leads to hallucinations easily. Our proposed BioKGBench is a dynamic benchmark that evaluates the capabilities of agents in utilizing external tools and knowledge retrieval, thereby addressing this gap.

**Agent Benchmark.** As agents are progressively applied across various domains, the urgency to construct corresponding benchmarks is escalating. Currently, the majority of benchmarks for evaluating agents adopt the approach of evaluating LLM-as-Agent (Liu et al., 2023c), linking LLMs to external frameworks to assess their performance on specific tasks. For instance, AgentBench (Liu et al., 2023c) is a general benchmark for evaluating an agent's reasoning and decision-making capabilities, SWE-bench (Jimenez et al., 2023) assesses an agent's proficiency in software engineering, and AgentClinic (Schmidgall et al., 2024) examines an agent's performance in a simulated clinical environment. However, a benchmark in AI Scientist perspective remains largely unexplored. Our benchmark originates from this perspective, taking the processing and understanding of large-scale data scenarios as the entry point, representing an initial attempt in this direction.

**Agent Integrating LLMs and KGs.** The collaborative use of LLM and KG has become one of the leading methodologies in contemporary agent design, aimed at alleviating uncertainties stemming from the intrinsic mechanisms of LLMs (Pan et al., 2024; Chen et al., 2023a; Yang et al., 2023c). This paradigm not only capitalizes on the generalization ability of LLMs but also employs KGs as an external, trustworthy, and structured data source, thereby achieving reasoning proficiency that strikingly emulates human intellect(Pan et al., 2024). For instance, StructGPT (Jiang et al., 2023) boosts an LLM's performance on general questions by tapping into the information from a supplied KG. Similarly, KG-Agent (Jiang et al., 2024b) leverages knowledge from KGs, synthesizing instruction data for fine-tuning an open-sourced LLM, thereby achieving competitive performance on general question-answering tasks. However, to our knowledge, while this paradigm has been widely applied to the general question-answering area, its potential remains untapped in the biomedical field. BKGAgent, hence, is poised to fill this gap.

## 3 BIOKGBENCH

Here, we present our benchmark in detail. As aforementioned, one key ability of "AI Scientists" is to understand domain knowledge. However, current LLM-driven agent systems inevitably suffer from hallucinations as a consequence of the statistical nature of LLMs along with the lack of scientific training data compared to data from daily scenarios. We notice that a recent trend in research is to use AI agents to leverage external tools to address these limitations (Bran et al., 2023; Bi et al., 2023).

Table 2: Comparison with existing well-known benchmarks.

| Benchmark | Domain | Dataset Composition | Multi-Turn | Environmental Interactio |
|---|---|---|---|---|
| MMLU (Hendrycks et al., 2020) | 57 subjects | QA | ✗ | ✗ |
| MATH (Hendrycks et al., 2021) | math | QA (including solution) | ✗ | ✗ |
| PubMedQA (Jin et al., 2019) | biomedical science | QA | ✗ | ✗ |
| SWE-bench (Jimenez et al., 2023) | software engineering | Issue text, codebase, gold patch, tests | ✗ | ✔ |
| MT-Bench (Zheng et al., 2023) | writing, math, knowledge | votes, conversations | ✔ | ✔ |
| AgentBench (Liu et al., 2023c) | LLM-as-Agent | 8 real-world tasks | ✔ | ✔ |
| **BioKGBench (ours)** | biomedical science, LLM-as-Agent | QA, **KG**QA, **KG**, literature | ✔ | ✔ |

Drawing inspiration from this, we design two atomic abilities to evaluate AI scientists, i) Knowledge Graph Question Answering (KGQA) aiming to address the hallucination issue by grounding the knowledge with structured knowledge graphs; and ii) Scientific Claim Verification (SCV) based on retrieved text from peer-reviewed research papers. In addition, we propose an encompassing task combining these

Table 1: Statistics of our BioKGBench.

| Task | Main Metrics | Scope | Data | | |
|---|---|---|---|---|---|
| | | | Dev | Test | All |
| KGQA | F1 | KG | 60 | 638 | 698 |
| SCV | Acc. | Text (T) | 120 | 1,265 | 1,385 |
| KGCheck | EM | KG + T | 20 | 205 | 225 |

two atomic abilities, to perform Knowledge Graph Checking (KGCheck) as shown in Figure 1. The motivation behind this stems from our interviews with experts from biomedical domains. Their answers to the question "What is the most expected AI agent you would like to use in your daily research?" often included an AI agent that helps in extensive literature review and claim verification. We report the statistics over the scopes of knowledge search, including knowledge graphs and academic literature, in Table 1 (Cf. Appendix A for more details). As shown in Table 2, compared to existing well-known benchmarks, BioKGBench features:

- **setting**: evaluating LLMs as agents through multi-round interactions with the environment to assess their ability to process and understand large-scale biomedical data.

- **data**: a diverse dataset of structured and unstructured data, allowing agents to derive knowledge from heterogeneous sources and make discoveries.

## 3.1 ATOMIC ABILITY

### 3.1.1 KNOWLEDGE GRAPH QUESTION ANSWERING

This atomic task in the benchmark is to evaluate the agents' ability to interact with structured Knowledge Graph Question Answering as a grounding of academic literature. Without loss of generality, we choose Clinical Knowledge Graph (CKG) (Santos et al., 2022) as the source of our data, which is one of the most popular large-scale knowledge graph databases in the biomedical domain. CKG is a knowledge graph database with data imported from diverse biomedical databases, aimed at streamlining automated knowledge discovery through the graph's extensive information.

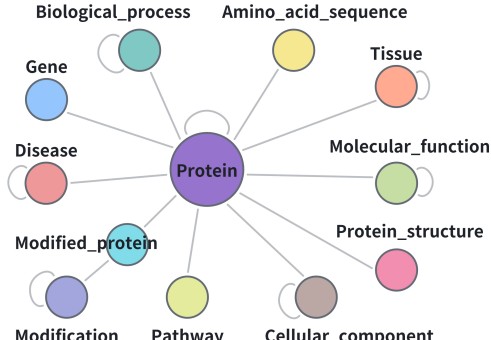

Figure 2: The sub-graph of the Clinical Knowledge Graph (CKG) retains 12 types of nodes and 18 kinds of relationships.

As the original database is unnecessarily large, we focus on a sub-graph to mitigate the challenge while preserving all relevant information. Starting from the origin of CKG—protein, we select the sub-graph to contain exactly 12 categories of biological entities, as indicated in Figure 2. Thus, the sub-graph consists of 484,955 entities (nodes) across 12 categories (Biologically defined) and 18,959,943 relationships (edges) of 18 types, with each type consisting of relationships between a unique pair of entity categories (Cf. Appendix A.2 for more details).

After the sub-graph is ready, we construct the question set for the Question Answering (QA) database in two steps. We first handcraft question templates by selecting biomedical fields and pinpointing

Table 3: Statics of three different reasoning types of KGQA dataset.

| Reasoning Type | Graph | Example Question | Question Types | % |
|---|---|---|---|---|
| **One-hop** | | What proteins does the protein O94842 act on? | 8 | 56.0 |
| **Multi-hop** | | What diseases are associated with the protein encoded by the gene KCNS1? | 4 | 28.7 |
| **Conjunction** | | Which pathway are the proteins P02778 and P25106 both annotated in? | 4 | 15.3 |

entities and relations in the CKG. Natural language questions were constructed in various formats, ensuring their accuracy through peer reviews and expert consultations. We then expanded our dataset with autogenerated questions by matching CKG data to constructed QA templates, resulting in the generation of 698 questions across three reasoning types and 16 question categories (refer to Table 3).

In this task, we outfit LLMs with a set of atomic KG-querying tools and ask them to answer biomedical questions by querying the provided KG. The responses will be compared with the gold answers and evaluated using the F1 score, where the gold answer to the input question is typically characterized by a set of KG entities. It is noteworthy that our KGQA is built upon a biomedical KG rather than a common sense KG, with the two adopting different data models. This difference is one of the reasons why KBQA methods cannot be directly applied (Cf. Appendix C.3 for more details). This task enables the development of assessing the robustness and tool learning ability of agents built upon various LLMs, and hopefully, it would aid in guiding agents to leverage the extensive biomedical knowledge within the KG, thereby propelling scientific discovery.

### 3.1.2 SCIENTIFIC CLAIM VERIFICATION

This task is designed to evaluate LLMs' understanding of unstructured text from research papers in a retrieval-augmented generation manner. Following the definition in (Wadden et al., 2020), the task is to identify evidence related to the claim from the research literature and give a verdict of "Support", "Refute", or "NEI" (Not Enough Information) based on it. We reconstruct two high-quality biomedical datasets, PubMedQA (Jin et al., 2019) and SciFact (Wadden et al., 2020), into one dataset for SCV, yielding a corpus constituted of abstracts derived from 5,664 scholarly articles, alongside a dataset comprising 1,385 biomedical claims, as shown in Table 4.

Table 4: Examples of reconstructed dataset for SCV, where data from PubMedQA is converted from QA to declarative claims. "NEI" stands for "Not Enough Information".

| Example Claim | Label | % |
|---|---|---|
| A deficiency of folate increases blood levels of homocysteine. | Support | 65.2 |
| Therapeutic anticoagulation in the trauma patient is safe. | Refute | 33.1 |
| Sternal fracture in growing children is a rare and often overlooked fracture. | NEI | 1.7 |

### 3.2 AGENT TASK

Building upon the atomic abilities, we propose a novel and comprehensive task, KGCheck. This task necessitates the initial application of the tool-query functionality to extract information from the KG. Subsequently, it employs the RAG approach or database access to procure evidence pertaining to the queried information, facilitating a determination of either "Support" or "Refute". This methodology enables agents to scrutinize the knowledge encapsulated within a large-scale KG, a venture of particular importance considering the prevalence of inaccuracies within numerous datasets, including prominent ones such as ImageNet (Deng et al., 2009).

For this task, we collect 225 high-quality annotated data, as illustrated in Table 5. Given the massive data encapsulated within KGs via triples, we delineate the inspection process into two distinct

Table 5: Four different checking types of KGCheck.

| Check Type | | Graph | % | Support: Refute |
|---|---|---|---|---|
| Node | Existence | ? | 20.0 | 71.0:29.0 |
| | Attribute | ? | 24.4 | |
| Triple | Existing | a —?→ b | 25.8 | 46.4:53.6 |
| | Potential | a --?--→ b | 29.8 | |

categories: single-node and triple-based. The single-node inspection is divided into node existence and attribute value assessments, while the triple inspection encompasses scenarios with and without edges between two nodes:

- **Existence**: We note that databases may excise entries during updates due to inaccuracies or redundancies, whereas KGs remain static post-construction, similar to LLMs in some respects. If nodes corresponding to obsolete entities persist in the KG, the label is "Refute"; if they are congruent with real-time updated external databases, the label is "Support".

- **Attribute**: Our KG is characterized by high information density, with each node and edge encapsulating numerous attribute values, which we scrutinize for accuracy and completeness.

- **Existing Relationship**: We check whether existing edges contradict information from external, real-time updated databases and literature. If external knowledge corroborates the relationship, the label is "Support"; conversely, it is "Refute".

- **Potential Relationship**: If a relationship is confirmed by databases or literature but is not represented in the KG, the label is "Refute"; otherwise, it is "Support".

Despite utilizing the latest databases (as of May 2024), we identified errors within the KG, evidenced by 96 "Refute" annotations. These data are valuable and provide scenarios for agents to comprehend knowledge from heterogeneous sources and make **discoveries**.

### 3.3 BKGAGENT: A SIMPLE BASELINE

We propose a **b**iomedical **k**nowledge-**g**raph **a**gent (BKGAgent), as shown in Figure 3. It's a multi-agent framework based on *lang-graph* (Chase, 2023), capable of retrieving information from knowledge graph and cross-validating its correctness with multiple information sources. Our framework is comprised of three agents: the team leader for the progress control, the KG agent for information retrieval from KG, and the validation agent for checking the correctness of the information from KG. This setup simulates the workflow of a human research team, where a leader supervises the assistants' work and makes the final decision based on their feedback. Additionally, the tool executor is solely responsible for executing functions, and is not based on LLMs.

Table 6: Comparison of capabilities for BKGAgent and other frameworks.

| Framework | MA | KGq | IR |
|---|---|---|---|
| HuggingGPT (Shen et al., 2023) | ✔ | ✗ | ✗ |
| OpenAgents (Xie et al., 2023) | ✔ | ✗ | ✗ |
| AgentVerse (Chen et al., 2023b) | ✔ | ✗ | ✗ |
| Xagent (Team, 2023) | ✔ | ✗ | ✗ |
| BabyAGI (Yoheinakajima) | ✔ | ✗ | ✗ |
| MedAgents (Tang et al., 2023c) | ✔ | ✗ | ✗ |
| gpt-researcher[1] | ✔ | ✗ | ✗ |
| BDAgent (Roohani et al., 2024) | ✗ | ✗ | ✔ |
| **BKGAgent(ours)** | ✔ | ✔ | ✔ |

BDAgent=BioDiscoveryAgent; MA=multi-agent; KGq=KG-query; IR=information retrieval

When a user assigns a task, the leader initially breaks down the task and announces the plan. Then the KG agent is activated to retrieve task-related information from the KG. This involves specifying the tool and its arguments to the tool executor, interpreting the tool result, and communicating it back to the leader. After that, the validation agent is called to verify the information with a workflow similar to that of the KG agent. Finally, the leader will draw a conclusion and return it to the user.

---

[1] https://gptr.dev/

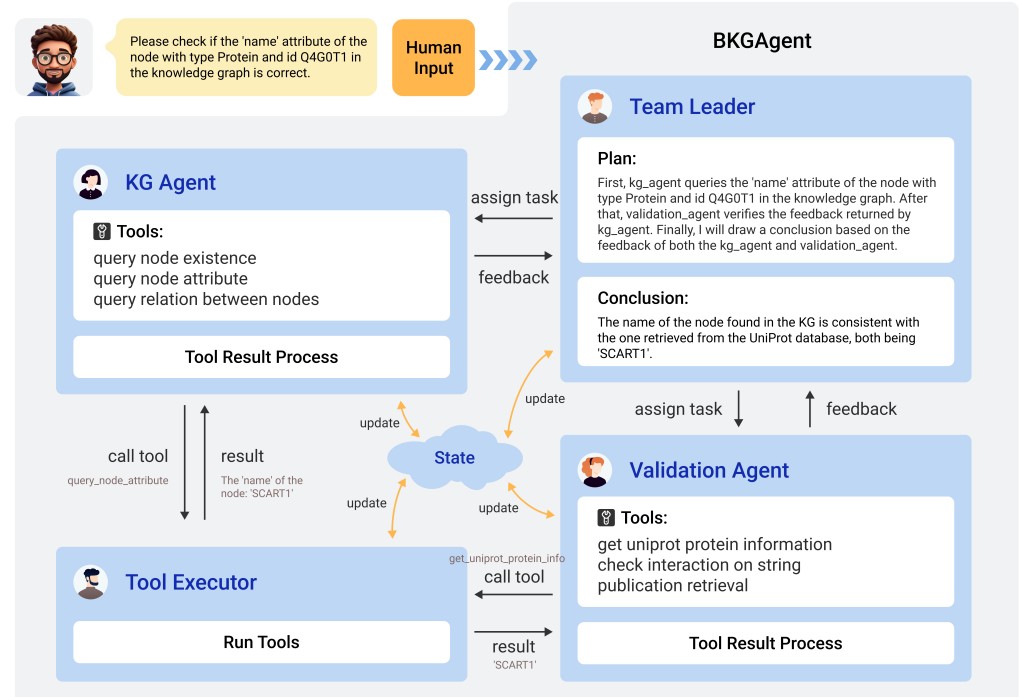

Figure 3: Framework of our BKGAgent.

BKGAgent possesses fundamental capabilities for grounding heterogeneous biomedical knowledge, including knowledge graph queries, database queries, and retrieval-augmented generation (RAG) of literature. In contrast, as illustrated in Table 6, many other frameworks struggle to achieve comparable effectiveness in biomedical information retrieval and verification due to their limited capacity to access knowledge graphs and biomedical data, as well as the unreliability of information sourced from the web.

## 4 EXPERIMENTS

### 4.1 MAIN RESULTS AND ANALYSIS: ATOMIC ABILITIES

**Metrics.** For KGQA, we adopt three metrics: F1, Exact Match (EM), and Executability. For SCV, we adopt three metrics: Accuracy, Right Quotes, and Error Rate. Specifically, "Executability" refers to the success rate of the agent providing an answer within 15 turns, "Right Quotes" indicates the success rate of retrieving matching text from the whole corpus through RAG, and "Error Rate" refers to the frequency with which the agent fails to make a verification.

The detailed experimental results of atomic abilities evaluation on LLMs are shown in Table 7, and we summarize our key findings as follows:

- **Disparity between open-source and commercial API models.** Commercial API models like GPT-4 and GLM-4 generally outperform open-source models in several key metrics. GPT-4, for example, consistently achieves higher scores in both KGQA and SCV tasks, highlighting the advantage of proprietary training techniques and larger computational resources.

- **Strong performance of open-source large models.** Some large OSS models, such as Llama-3-70B-Instruct, perform competitively, sometimes surpassing API models in specific metrics. Llama-3-70B-Instruct, in particular, excels in KGQA executability, suggesting that optimized training can enable open-source models to rival or exceed commercial counterparts.

- **Model parameters do not always correlate with better performance.** In the OSS (Medium) and OSS (Small) categories, smaller models like Llama-3-8B-Instruct sometimes outperform larger models like Qwen1.5-32B-Chat in SCV tasks, indicating that model architecture, training data

Table 7: Test set (standard) results of two easy tasks: KGQA, SCV. **Bold/**underline and red/blue indicate the best and second in the subgroup and overall.

| LLM Type | Models | KGQA | | | SCV | | |
|---|---|---|---|---|---|---|---|
| | | F1 | EM | Executability | Accuracy | Right Quotes | Error |
| API | GPT-4 (OpenAI, 2023a) | 81.8 | 79.2 | **88.4** | 83.9 | 87.7 | **0.4** |
| | GLM-4 (Du et al., 2022) | 72.4 | 70.4 | 82.7 | 86.9 | 86.5 | 0.6 |
| OSS (Large) | Qwen1.5-72B-Chat (Bai et al., 2023) | 74.7 | 72.2 | 96.1 | 85.7 | 83.3 | 0.1 |
| | Llama-3-70B-Instruct (AI@Meta, 2024) | **80.7** | **77.8** | **97.0** | **85.9** | **86.6** | 0.2 |
| | DeepSeek-LLM-67B-Chat (Bi et al., 2024) | 69.6 | 66.8 | 86.3 | 76.6 | 82.6 | 0.4 |
| OSS (Medium) | Qwen1.5-32B-Chat (Bai et al., 2023) | 64.6 | **62.1** | **83.0** | 79.7 | **83.0** | 0.4 |
| | Qwen1.5-14B-Chat (Bai et al., 2023) | **66.0** | 61.6 | 78.7 | 66.1 | 67.4 | 0.2 |
| | Baichuan2-13B-Chat (Yang et al., 2023a) | 43.7 | 42.0 | 82.2 | 26.3 | 35.8 | 33.6 |
| OSS (Small) | Llama-3-8B-Instruct (AI@Meta, 2024) | **54.7** | **51.3** | **84.8** | 78.5 | **83.3** | **0.5** |
| | Qwen1.5-7B-chat (Bai et al., 2023) | 44.5 | 40.3 | 77.9 | 72.5 | 39.1 | 2.2 |
| OSS (MoE) | Mixtral-8x7B-Instruct-v0.1 (Jiang et al., 2024a) | **70.1** | **67.9** | **84.7** | 77.8 | **82.5** | 2.3 |
| | Starling-LM-alpha-8x7B-MoE-GPTQ (Zhu et al., 2023) | 12.4 | 10.9 | 30.7 | 55.0 | 56.2 | 0.1 |
| | Qwen1.5-MoE-A2.7B-Chat (Bai et al., 2023) | 28.7 | 26.7 | 71.9 | 55.0 | 57.8 | 3.0 |

quality, and fine-tuning strategies significantly impact performance. Notably, Qwen1.5-14B-Chat outperforms Qwen1.5-32B-Chat in KGQA, suggesting the latter's pre-training may be insufficient.

- **Domain-specific models lack transferability.** DeepSeek-LLM-67B-Chat excells in mathematical problems (Bi et al., 2024), but underperforms in biomedical-related tasks, highlighting its lack of cross-domain transferability. This suggests that specialization in one area may compromise generalizability.

- **Inconsistent performance of MoE models.** While Mixtral-8x7B-Instruct-v0.1 performs well in both KGQA and SCV tasks, other MoE models like Starling-LM-alpha-8x7B-MoE-GPTQ and Qwen1.5-MoE-A2.7B-Chat show significantly lower scores. This inconsistency suggests that the effectiveness of MoE models heavily depends on the implementation and integration of the expert models. Additionally, Mixtral-8x7B-Instruct-v0.1, though strong in main metrics, struggles with controlling response format, indicating that individual expert models still require improvement.

- **Biomedical knowledge embedded in model parameters.** The new metric "Right Quotes" for SCV assesses the alignment of retrieved quotes with ground truth evidence. Some models, such as GLM-4, Qwen1.5-72B-Chat, and Qwen1.5-7B-Chat, exhibit higher accuracy metrics than "Right Quotes" metrics. This suggests these models can accurately assess input claims even without sufficient literature evidence, indicating they possess specialized biomedical knowledge.

**Further Analysis.** We also conduct an ablation experiment on three scopes of RAG, as shown in Figure 4, where 'all' refers to the abstract of 5,664 articles, 'partial' denotes the 1,888 abstracts containing ground truth evidence of claims, and 'match' corresponds to the abstracts of the ground truth evidence for the claims. Interestingly, we observe an unexpected phenomenon where the model's performance in the 'match' setting only increases in terms of the right quotes metric, while the accuracy metric decreases. In the 'all' setting, we initially anticipated interference from irrelevant literature, but the accuracy

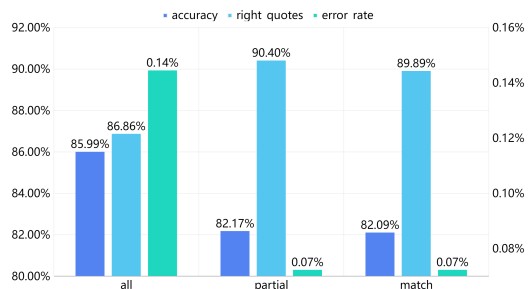

Figure 4: Llama-3-70B-Instruct's performance in RAG across different scopes of literature.

metric instead improved. This suggests that there is a potential connection among the extensive literature, where large models exhibit a form of **"analogical reasoning"**. This provides us with insights for conducting extensive literature research in simulating human scientific research.

## 4.2 MAIN RESULTS AND ANALYSIS: BKGAGENT

**Evaluation Setup.** As mentioned in 3.3, most agent frameworks fail in KGCheck, which further highlights that KGCheck is a novel and challenging task. It requires agents to first query the knowledge graph, followed by verification through database searches or RAG of literature. Consequently, agents lacking capabilities for KG querying or information retrieval verification cannot complete this task. Therefore, for agents capable of querying knowledge graphs, we selected KG-Agent from (Liu et al., 2023c) as representative; for general agent frameworks, we chose the most prominent ones,

Table 8: The performance of different agents built on GPT-4o (Hel) in executing the KGCheck task. **Bold/** underline indicate the best and second, respectively. All scores are on a percentage scale.

| Agent | Process | | | | | | Result |
|---|---|---|---|---|---|---|---|
| | Understanding | Reasoning | Efficiency | KG process | Information retrieval | Average | EM |
| **Baselines** | | | | | | | |
| AgentBench-KG agent | 81.0 | 81.0 | 81.0 | 77.1 | 97.1 | 83.4 | 56.6 |
| AutoGPT | **99.0** | 78.0 | 84.4 | 96.6 | 59.5 | 83.5 | 39.5 |
| AutoGen | 95.6 | 45.9 | 78.0 | 28.8 | 30.7 | 55.8 | 30.2 |
| **Ours** | | | | | | | |
| AgentBench-KG agent w/ our tools | 98.5 | 84.4 | 85.9 | 97.6 | 91.7 | 91.6 | 68.8 |
| AutoGPT w/ our tools | **99.0** | 97.0 | **98.5** | 99.0 | 99.0 | 98.5 | 75.1 |
| AutoGen w/ our tools | **99.0** | **99.0** | 98.0 | 99.0 | **100.0** | **99.0** | 77.1 |
| **BKGAgent (ours)** | 89.8 | 94.1 | 95.1 | **100.0** | 95.1 | 94.8 | **78.0** |

AutoGen (Wu et al., 2023a) and AutoGPT[2], along with their three improved versions, as well as our BKGAgent, for comparison, as shown in Table 8[3]. Both the final results and process are considered for a more robust evaluation. Since the ground truth is either "support" or "refute", we use Exact Match (EM) as the metric for the final result. For the process, we employ Qwen2-72B[4] to score based on five criteria: (1) Understanding: whether the agent clearly understood the task and the purpose of the given tool. (2) Reasoning: whether the agent arrived at the final answer through sufficient evidence and reasoning, rather than simply providing random answers or guessing. (3) Efficiency: whether the agent efficiently solved the problem without unnecessary discussion on unrelated topics. (4) KG Process: whether the agent queried the knowledge graph during the task. (5) Information Retrieval: whether the agent retrieved information from external knowledge sources in some way during the check. To align the judgments made by LLMs closely with those of humans, we collect 10 agent histories along with human-annotated scores (on a 5-point scale) and prompt the LLM to produce scores that closely resemble human ratings. We take EM as the main metric, while process scores serve as supplementary metrics.

**Agent Comparison.** Table 8 shows that BKGAgent outperforms the other agents. KG-Agent achieved an accuracy of only 56.6%, roughly equivalent to random guessing. This aligns with our expectations, as while it can accurately query information from the knowledge graph, it lacks access to reliable external knowledge sources for verification, leading to hallucinations in the large model's guesses. Notably, the final accuracy of Vanilla AutoGen and AutoGPT is quite low, at just over 30%. This underscores the importance of integrating general capabilities with specialized tools to enhance agent performance. Their performance suffers because they are general frameworks that rely on some general capabilities like programming and web searches, which are not robust enough, often resulting in execution failures due to poor code quality. Consequently, they cannot provide answers within the limited interaction turns. Thus, we improved KG-Agent, AutoGen, and AutoGPT by equipping them with tools including KG querying and RAG. We also designed prompts to teach them how to utilize these tools. As a result, KG-Agent w/ our tools, AutoGen w/ our tools, and AutoGPT w/ our tools demonstrate significant improvements, highlighting that the integration of general capabilities with specialized tools enhances the robustness of agent performance.

**Case Study of BKGAgent.** While the behavior of the assistant agents in BKGAgent can be modified by the leader's instruction, the leader itself lacks action-related feedback from others, meaning that a bad decision made by the leader may lead to a catastrophe. We found four common error cases induced by the leader, as shown in Figure 6. Among these cases, the leader either fails to give effective instructions to team members, becomes trapped in repeated self-talks, or attempts to perform the tasks that are meant for the assistants. Thus, we can conclude that the performance of the leader agent significantly impacts team behavior. Furthermore, compared to the other agent frameworks in Table 8, BKGAgent employs the maximum number of three agents. However, in our additional case studies (see Appendix D.3), we found that an increase in the number of agents also leads to higher communication costs and a decrease in fault tolerance. They frequently experience task failures due to being lost in conversation.

---

[2] https://news.agpt.co/

[3] Other agent frameworks that can perform information retrieval but cannot query knowledge graphs are unable to execute the task from the outset, resulting in scores of zero; therefore, we omit their evaluation.

[4] https://huggingface.co/Qwen/Qwen2-72B-Instruct

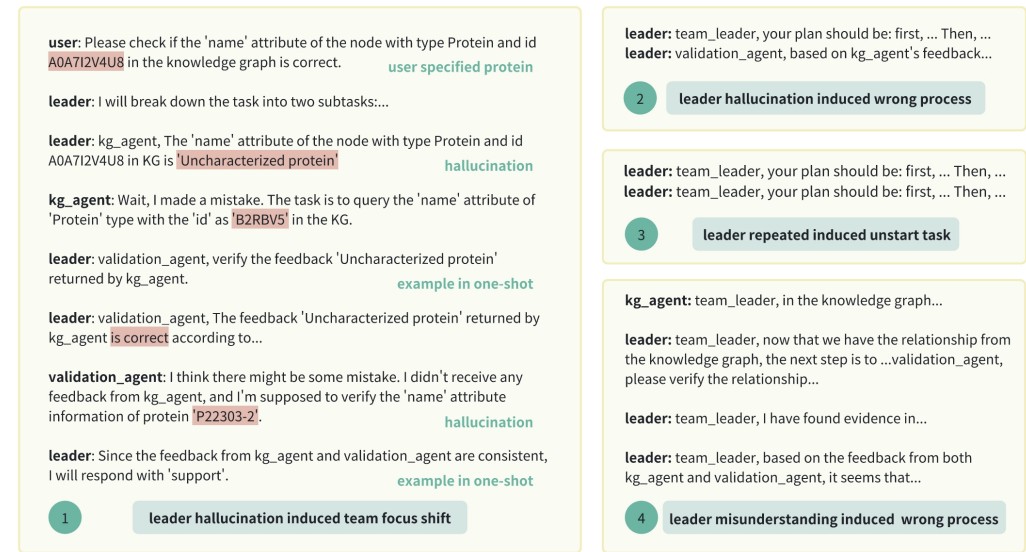

Figure 6: **Error analysis.** Here, we show a failure case due to a leader's various mistakes: the hallucination of the leader misleading the later task or using the wrong process, the leader producing unnecessary repeated tasks and misunderstanding leads to the wrong process.

**Impact of Agent Number on System Performance: A Further Analysis.** By equipping agents with identical capabilities (e.g., KG querying, database querying, and RAG of literature), we compare the performance of systems with 1, 2, and 3 agents (Cf. Figure 5). Our BKGAgent achieves a 20% higher recall rate of errors (i.e., the ability to correctly identify errors in the KG) than the second-best system, demonstrating its strong performance. However, we also note that the advantage of using 3 agents in terms of EM is not significant. While adding agents slightly improves performance through collaboration, it also increases communication costs and complexities, leading to diminishing returns. In contrast, the application of effective algorithms, such as ReACT (Yao et al., 2022), can yield more significant improvements, as evidenced by the comparison between AgentBench-KG Agent w/ our tools and AutoGPT w/ our tools.

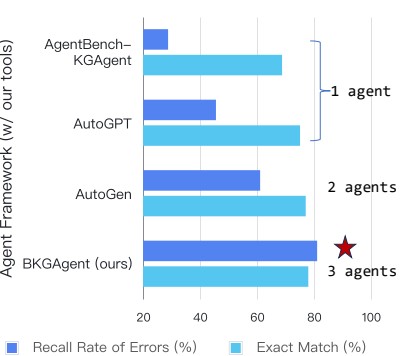

Figure 5: Comparison of the performance of agent frameworks with 1/2/3 agents.

## 5 CONCLUSION

We present BioKGBench, an interactive benchmark that encompasses the KGCheck task with two atomic capabilities for assessment: KGQA and SCV. KGCheck offers agents a valuable scenario for detecting knowledge hallucination within large-scale data, akin to the experience of researchers making discoveries amidst voluminous literature in the real world. We conduct evaluations of these two atomic capabilities across 13 LLMs and select GPT-4o, to construct BKGAgent——a multi-agent system serving as the baseline. Comparisons with existing general and biomedical agents revealed their poor performance due to the absence of certain process capabilities, thereby demonstrating the challenging nature of our benchmark. We expect BioKGBench to serve as a valuable endeavor towards paving the path for biomedical agents to become AI scientists.

**Limitations and Future Work.** In KGCheck, we guide agents to identify knowledge-based errors within the KG by providing them with specific instructions. This process involves atomic-level inspections from single nodes to triples, which agents could potentially implement autonomously. Future work will explore how agents can autonomously conduct real-time error detection in large datasets by leveraging logic rules and prior knowledge.

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

## A    DATASHEET

Here, we provide a detailed description of our benchmark dataset, following the guidelines of the "Datasheet for Datasets" (Gebru et al., 2021).

### A.1    MOTIVATION

Our benchmark dataset was created to address the lack of benchmarks for evaluating biomedical agents from the perspective of an "AI scientists". In (Gao et al., 2024), it is stated that "AI scientists" can be realized as AI agents supported by humans, LLMs, ML models, and other tools like experimental platforms that cooperate to solve complex tasks. However, the current evaluation methods for biomedical agents remain unexplored, limited to simple question-answering tasks, which not only fail to avoid the hallucination problem inherent in solely relying on LLMs but also do not assess agents' abilities to utilize external tools and knowledge bases. Our proposed benchmark fills this gap by designing tasks ranging from easy to hard, based on two atomic capabilities: tool-query and memory-RAG. These tasks evaluate the agents' ability to leverage external support, including external knowledge and tools, when handling large-scale and multi-modal data. Moreover, Our data collected for KGCheck, the most challenging task, provides scenarios for agents to comprehend knowledge from heterogeneous sources and make discoveries.

The conception and construction of this dataset were jointly completed by the biomedical experts and AI researchers listed in the author list.

### A.2    COMPOSITION

We provide the necessary data for constructing the knowledge graph, literature for RAG, as well as the development and test data for KGQA, SCV, and KGCheck, where knowledge graph and literature are external knowledge sources provided for agent.

The knowledge graph is derived from a subset of Clinical Knowledge Graph (CKG) (Santos et al., 2022). We specifically retain twelve key node types to ensure there is no loss of generality: Protein, Biological process (BP), Molecular function (MF), Cellular component (CC), Amino acid sequence, Tissue, Protein structure, Pathway, Modified protein, Modification, Disease, and Gene. The statistics of the triples in our knowledge graph are presented in Table 9. Detailed information stored in our knowledge graph is shown in Table 10.

Table 9: The data statistics of our knowledge graph drawn from CKG.

| Head Node | Tail Node | Relation | Number |
| --- | --- | --- | --- |
| Protein | Protein_structure | HAS_STRUCTURE | 271,512 |
| | Amino_acid_sequence | HAS_SEQUENCE | 20,598 |
| | Cellular_component | ASSOCIATED_WITH | 3,796,383 |
| | Tissue | ASSOCIATED_WITH | 7,117,321 |
| | Disease | ASSOCIATED_WITH | 5,882,437 |
| | Molecular_function | ASSOCIATED_WITH | 85,013 |
| | Biological_process | ASSOCIATED_WITH | 153,219 |
| | Protein | ACTS_ON | 985,376 |
| | Pathway | ANNOTATED_IN_PATHWAY | 357,739 |
| | Protein | CURATED_INTERACTS_WITH | 3,448 |
| | Modified_protein | HAS_MODIFIED_SITE | 4,498 |
| Disease | Disease | HAS_PARENT | 16,058 |
| Modified_protein | Protein | IS_SUBSTRATE_OF | 6,633 |
| | Modification | HAS_MODIFICATION | 4,559 |
| Gene | Protein | TRANSLATED_INTO | 179,854 |
| Biological_process | Biological_process | HAS_PARENT | 49,081 |
| Molecular_function | Molecular_function | HAS_PARENT | 13,659 |

Besides the knowledge graph, literature also serves as an external source of knowledge. We provide a corpus of 5,664 abstracts (under ODC-By 1.0) for SCV and 51 full papers for KGCheck. The 5,664

Table 10: Details of the information stored in the nodes of our knowledge graph.

| Entity Type | Content | Example |
|---|---|---|
| Protein | name, id, accession, synonyms | {'name': 'PLEKHG6', 'id': 'Q3KR16', 'accession': 'PKHG6_HUMAN', 'synonyms': ['PKHG6_HUMAN', 'PLEKHG6', '9606.ENSP00000380185', 'ENSG00000008323'], 'taxid': '9606'} |
| Disease | name, description, id(DOID), type, synonyms | {'synonyms': ['sulfamethoxazole allergy', 'SMX allergy', 'SMZ allergy', 'sulphamethoxazole allergy'], 'name': 'sulfamethoxazole allergy', 'description': 'A drug allergy that has_allergic_trigger sulfamethoxazole. [url:https://www.ncbi.nlm.nih.gov/pubmed/7602118]', 'id': 'DOID:0040016', 'type': '-26'} |
| Protein structure | link, id, source | {'link': http://www.rcsb.org/structure/6XWD, 'id': '6XWD', 'source': 'Uniprot'} |
| Amino acid sequence | sequence, header, source, id, size | {'sequence': 'LRGAAGRLGGGLLVL', 'size': '15', 'header': 'tr|A0A0A0MTA2|A0A0A0MTA2_HUMAN', 'source': 'UniProt', 'id': 'A0A0A0MTA2'} |
| Cellular component | name, description, id, type, synonyms | {'name': 'Golgi membrane', 'description': 'The lipid bilayer surrounding any of the compartments of the Golgi apparatus. [GOC:mah]', 'id': 'GO:0000139', 'type': '-21', 'synonyms': ['Golgi membrane']} |
| Molecular function | name, description, id(GO), type, synonyms | {'name': 'polymeric immunoglobulin binding', 'description': 'Interacting selectively and non-covalently with a J-chain-containing polymeric immunoglobulin of the IgA or IgM isotypes. [GOC:add, ISBN:0781735149]', 'id': 'GO:0001790', 'type': '-21', 'synonyms': ['polymeric immunoglobulin binding']} |
| Biological process | name, description, id(GO), type, synonyms | {'synonyms': ['mitochondrion inheritance'], 'name': 'mitochondrion inheritance', 'description': 'The distribution of mitochondria, including the mitochondrial genome, into daughter cells after mitosis or meiosis, mediated by interactions between mitochondria and the cytoskeleton. [GOC:mcc, PMID:10873824, PMID:11389764]', 'id': 'GO:0000001', 'type': '-21'} |
| Pathway | name, description, linkout, id, source | {'name': 'Antigen processing: Ubiquitination & Proteasome degradation', 'description': 'Antigen processing: Ubiquitination & Proteasome degradation', 'linkout': https://reactome.org/PathwayBrowser/#/R-HSA-983168, 'id': 'R-HSA-983168', 'source': 'Reactome'} |
| Tissue | name, description, id, type, synonyms | {'name': 'stratum basale', 'description': 'The deepest layer, as of the epidermis or the endometrium. In the epidermis it is a single layer of cells. In the endometrium it provides the regenerative tissue after menstrual loss of the functional layer. [Dorlands_Medical_Dictionary:MerckMedicus]', 'id': 'BTO:0004680', 'type': '-25', 'synonyms': ['stratum basale', 'basal layer']} |
| Modified protein | sequence_window, protein, position, source, id, residue | {'sequence_window': 'MEPAPARsPRPQQDP', 'protein': 'P29590', 'position': '8', 'source': 'SIGNOR', 'id': 'P29590_S8-p', 'residue': 'S'} |
| Modification | synonyms, name, description, id, type | {'synonyms': ['Unimod', 'Source: "none"'], 'name': 'Unimod', 'description': 'Entry from Unimod. [PubMed:18688235]', 'id': 'MOD:00003', 'type': '-41'} |
| Gene | taxid, synonyms, name, id, family | {'taxid': '9606', 'synonyms': ['54843', 'ENSG00000137501', 'OTTHUMG00000166977', 'uc010rti.4', 'AJ303364'], 'name': 'synaptotagmin like 2', 'id': 'SYTL2', 'family': '"Protein phosphatase 1 regulatory subunits|Synaptotagmin like tandem C2 proteins"'} |

abstracts are sourced from existing datasets SciFact (Wadden et al., 2020) (under CC BY 4.0) and PubMedQA (Jin et al., 2019) (under MIT license), while the 51 full papers, all of which are open access, were selected by experts based on entries in the IntAct (Orchard et al., 2014) database and CKG. Table 11 summarizes the sources of the abstracts, and Figure 7 describes the literature with more details. Table 12 summarizes the sources of the 51 full papers, and Figure 8 provides more details about the literature.

Table 11: The 5,664 papers come from 1,484 journals. Due to space limitations, we only list the names of journals with an IF greater than 70 and use 'others' to represent journals with an IF less than 70.

| Journal | Count |
|---|---|
| Nature reviews. Microbiology | 3 |
| CA: a cancer journal for clinicians | 3 |
| The Lancet. Infectious diseases | 3 |
| Nature reviews. Drug discovery | 5 |
| Nature reviews. Molecular cell biology | 14 |
| Nature reviews. Immunology | 21 |
| Lancet (London, England) | 46 |
| The New England journal of medicine | 46 |
| BMJ (Clinical research ed.) | 90 |
| JAMA | 113 |
| Nature medicine | 138 |
| others | 5182 |
| Total | 5664 |

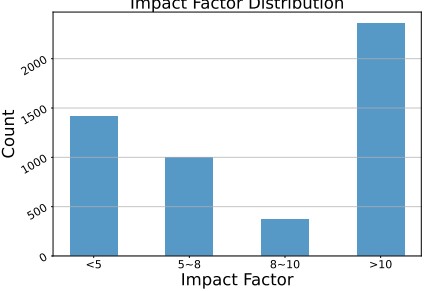

(a) Distribution of literature IF (Impact Factor)

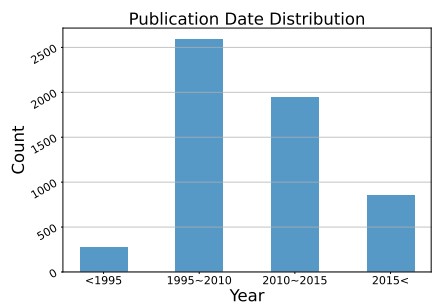

(b) Distribution of literature publication dates

Figure 7: Specific information of the 5,664 abstracts

For evaluation, we carefully collected 698, 1385, and 225 instances for KGQA, SCV, and KGCheck respectively. These datasets are split into development (dev) and test sets at an approximate ratio of 1:10. The dev data is intended for users to debug and fine-tune their evaluation code, while the test data is reserved for the final assessment. Each instance includes both input and output (ground truth answer or label) pairs, with additional information to make the data easier to understand. The dataset for SCV is reconstructed from well-known existing datasets SciFact and PubMedQA, while the rest is self-contained. The dataset represents a carefully selected sample of instances from a larger set, ensuring a comprehensive and representative coverage of the key aspects.

## A.3 COLLECTION PROCESS

Biomedical experts and AI researchers listed in the author list were invloved in the data collection process. The collection process for different tasks varies:

**KGQA.** The collection process can be summarized into two steps: manually constructing question templates and automatically generating questions:

Table 12: Journal distribution of the 51 full papers

| Journal | Count |
|---|---|
| Brain research | 1 |
| Molecular & cellular proteomics : MCP | 1 |
| Aging cell | 1 |
| Cell reports | 1 |
| PloS one | 1 |
| Genes to cells : devoted to molecular & cellular mechanisms | 1 |
| EMBO reports | 1 |
| IUBMB life | 1 |
| The Journal of allergy and clinical immunology | 1 |
| The EMBO journal | 1 |
| Open biology | 1 |
| Nature communications | 1 |
| Pigment cell & melanoma research | 1 |
| Molecular biology of the cell | 1 |
| Mobile DNA | 1 |
| Journal of molecular biology | 1 |
| Nutrients | 1 |
| Biological research for nursing | 1 |
| Genes & development | 1 |
| Developmental cell | 1 |
| Bone | 1 |
| Cancers | 1 |
| Animals : an open access journal from MDPI | 1 |
| Nucleic acids research | 2 |
| Molecular cell | 2 |
| Scientific reports | 3 |
| Proceedings of the National Academy of Sciences of the United States of America | 3 |
| Molecular and cellular biology | 4 |
| Cell | 4 |
| The Journal of biological chemistry | 10 |
| Total | 51 |

- **Workflow for the Handcrafted Question Templates:** The process commenced by selecting specific biomedical research fields and identifying relevant entity types and relationships from our knowledge graph. We defined various types of natural language questions, including one-hop questions, multi-hop questions, and conjunction questions (involving multiple entities). For each question, we created corresponding queries in both human-readable and machine-readable formats. These questions and queries, along with their associated metadata, such as question type and query structure, underwent rigorous peer reviews to ensure syntactic and semantic correctness.

- **Workflow for the Auto-generated Questions and Answers:** In the expansion of our benchmark, we initiated the process with the creation of auto-generated question templates. For instance, we used handcrafted question templates like "Which pathway are the proteins <Protein1> and <Protein2> both annotated in?" and then scoured our knowledge graph for data that fit the criteria to formulate both questions and answers, thereby augmenting the size of our dataset. This comprehensive dataset enables the development of assessing the robustness and accuracy of various LLM agents, providing a comprehensive benchmark that contributes to the advancement of the field with extensive biomedical knowledge.

**SCV.** We combine two high-quality biomedical datasets, PubMedQA and SciFact, into a single dataset for SCV. This results in a corpus consisting of abstracts from 5,664 scholarly articles and a dataset containing 1,385 biomedical claims. To further ensure consistency, we conducted secondary

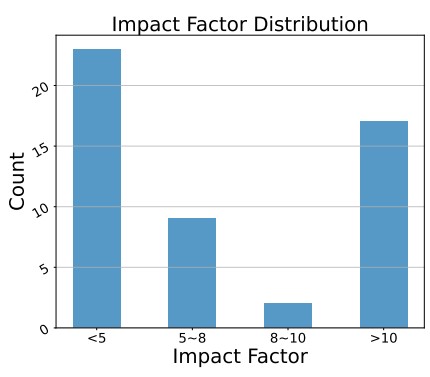 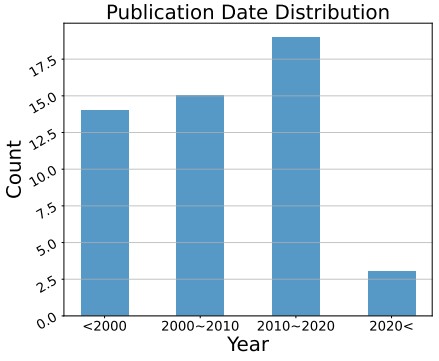

(a) Distribution of literature IF (Impact Factor)     (b) Distribution of literature publication dates

Figure 8: Specific information of the 51 full papers

verification on this expert-annotated dataset using Qwen1.5-72B (Bai et al., 2023), confirming the claims are conflict-free.

**KGCheck.** Considering the characteristics of the knowledge graph, we decompose the approach to checking the knowledge graph into two atomic-level checks: nodes and triples. Further, we subdivide these into whether a node should exist in the knowledge graph, whether the information stored in the node is correct, whether the relationship between two connected nodes in the knowledge graph truly exists, and whether there is a potential relationship between two nodes that are not connected by an edge. To collect this data, we selected well-maintained external knowledge sources such as the UniProt database, the IntAct database, and literature. We cross-verified the information in our knowledge graph with these reliable sources, labeling mutually corroborative data as 'support' and data that contradicts the external reliable sources as 'refute'. Specifically, for the data collection to check nodes, we review some update information from databases, such as entries removed due to errors or entries with updated information. Based on this information, we used Cypher queries to check our knowledge graph and label the data accordingly. For checking triple relationships, we sampled some triples from our knowledge graph where two nodes were either related or unrelated. We then queried the CKG to obtain literature that documents the entities represented by both nodes. We collected the literature annotated in the database and had experts read these documents to label the relationships of the triples in the CKG. As a result, we obtained 225 high-quality annotated data.

### A.4 USES

The dataset has not been used for any tasks yet. Currently, we have not identified any tasks that are not permitted to use our dataset.

The way we collect question and answer pairs can be referenced to expand more KGQA data, whether on our knowledge graph or new knowledge graphs. Additionally, our approach to collecting data for KGCheck provides insights into identifying errors in these large knowledge graphs, which is very helpful for subsequent error correction and data updates.

## B BREAKDOWN RESULTS

### B.1 KGQA

We conducted a more detailed evaluation of LLMs' performance on the KGQA task based on the question types: one-hop, multi-hop, and conjunction. The evaluation metrics used were F1 and executability, as shown in Table 13. We find that although API-based commercial LLMs and large-scale open-source models generally perform well on overall metrics, when breaking down the KGQA task by question type, some medium-scale models perform better on certain metrics. For instance, Qwen1.5-14B-Chat exhibits higher executability on more challenging multi-hop and conjunction

Table 13: KGQA Test set (standard) results by question type: one-hop, multi-hop, and conjunction. **Bold**/underline and red/blue indicate the best and second in the subgroup and overall.

| LLM Type | Models | F1 | | | executability | | |
|---|---|---|---|---|---|---|---|
| | | one-hop | multi-hop | conjunction | one-hop | multi-hop | conjunction |
| API | GPT-4 | **87.2** | **73.7** | 77.4 | **88.0** | 90.0 | 86.9 |
| | GLM-4 | 76.0 | 73.0 | 58.0 | 82.9 | 90.0 | 68.2 |
| OSS (Large) | Qwen1.5-72B-Chat | 76.3 | **73.4** | 71.4 | **99.7** | 94.0 | 86.9 |
| | Llama-3-70B-Instruct | **83.6** | 72.5 | **85.1** | 95.7 | 98.5 | **99.1** |
| | DeepSeek-LLM-67B-Chat | 80.6 | 61.8 | 44.1 | 88.5 | 90.5 | 70.1 |
| OSS (Medium) | Qwen1.5-32B-Chat | **67.3** | 63.2 | 57.0 | **87.2** | 84.5 | 64.5 |
| | Qwen1.5-14B-Chat | 63.7 | **70.5** | **65.7** | 67.5 | **95.5** | **87.9** |
| | Baichuan2-13B-Chat | 64.9 | 20.4 | 9.8 | 81.8 | 91.5 | 66.4 |
| OSS (Small) | Llama-3-8B-Instruct | **59.2** | **66.4** | 16.5 | **90.8** | 66.4 | 68.2 |
| | Qwen1.5-7B-chat | 55.7 | 32.1 | **26.4** | 80.3 | 79.0 | 67.3 |
| OSS (MoE) | Mixtral-8x7B-Instruct-v0.1 | **80.3** | **68.4** | **35.9** | 90.5 | 91.5 | 50.5 |
| | Starling-LM-alpha-8x7B-MoE-GPTQ | 6.2 | 25.0 | 11.7 | 12.0 | 57.5 | 48.6 |
| | Qwen1.5-MoE-A2.7B-Chat | 38.2 | 20.2 | 9.7 | **94.4** | 45.0 | 40.2 |

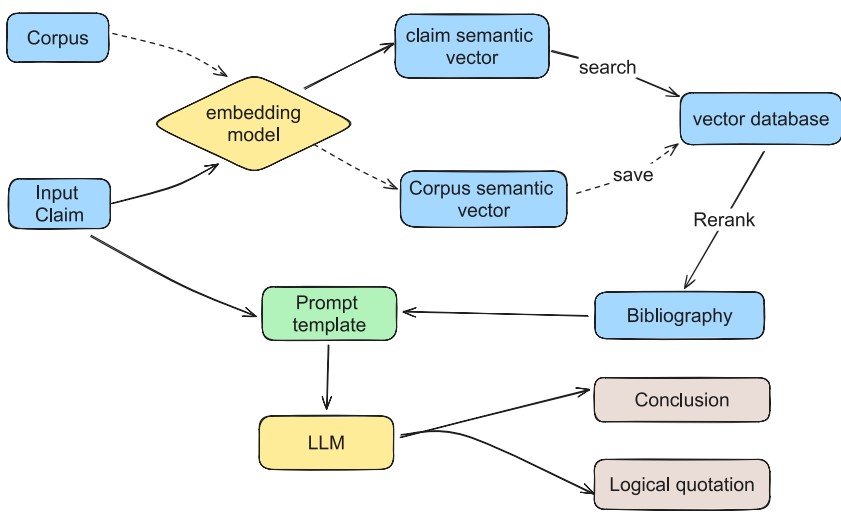

Figure 9: The pipeline of RAG.

types of questions, although its F1 score is not high. In terms of the executability metric, open-source models seem to outperform API-based commercial LLMs. This may be because API-based LLMs are more cautious in determining whether an answer has been obtained, tending to conclude the interaction and return an answer only after confirming its correctness.

## B.2   SCV

As shown in Figure 9, this is the process we followed when performing the SCV task using RAG. In the main text, we observed an interesting phenomenon where expanding the RAG scope improved accuracy. To ensure that this result was not due to the idiosyncratic performance differences of a single model, we conducted the same experiment on another model, as shown in Table 14. It can be observed that the accuracy of both models in the SCV task increased with the expansion of the RAG scope, although the right quotes metric was the lowest across the three settings when performing RAG at the maximum scope. This experimental result further demonstrates that this interesting phenomenon is not due to model-specific characteristics.

Table 14: Supplementary Experiments on the Scope of RAG, where 'all' refers to the abstract of 5,664 articles, 'partial' denotes the 1,888 abstracts containing ground truth evidence of claims, and 'match' corresponds to the abstracts of the ground truth evidence for the claims. **Bold**/underline indicate the best and suboptimal.

| Corpus | Qwen1.5-72B-Chat | | | Llama-3-70B-Instruct | | |
|---|---|---|---|---|---|---|
| | accuracy | right quotes | error | accuracy | right quotes | error |
| all | **86.2** | 82.1 | **0.1** | **86.0** | 86.9 | 0.2 |
| partial | 84.4 | 88.1 | **0.1** | 82.2 | **90.4** | **0.1** |
| match | 84.3 | **88.2** | **0.1** | 82.1 | 89.9 | **0.1** |

## B.3 KGCHECK

### B.3.1 SINGLE AGENT

We develop a single agent based on KG-Agent from (Liu et al., 2023c), shown in Figure 10, and evaluate LLMs as agents by replacing the LLM with a specific model. We set up a Single Agent (see Figure 1 and Table 2) and compared it with the BKGAgent, which is a multi-agent system. The results are shown in Table 15.

Table 15: Performance of single agent on the KGCheck task. **Bold**/underline indicate the best and second.

| LLM Type | Models | KGCheck | |
|---|---|---|---|
| | | EM | Executability |
| API | GPT-4o | 68.8 | 98.0 |
| | GLM-4-0520 | 51.7 | 96.1 |
| OSS | Qwen2-72B-Instruct | **82.9** | **100.0** |
| | Qwen1.5-72B-Chat | 43.4 | **100.0** |
| | Llama-3-70B-Instruct | 76.1 | 96.6 |
| | Llama-3.1-8B-Instruct | 44.4 | 87.8 |
| | Mixtral-8x7B-Instruct-v0.1 | 57.1 | 98.0 |

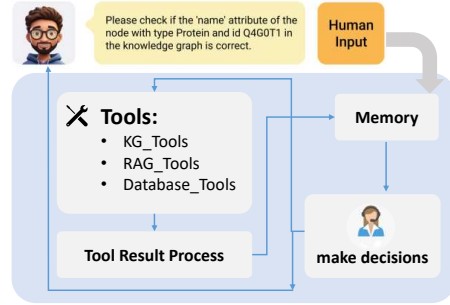

Figure 10: Single agent for KGCheck task.

### B.3.2 BKGAGENT

We exhibit our BKGAgent performance on KGCheck tasks based on the data source for verification(i.e. web database KGCheck and publication database KGCheck) in the main body for clarity. However, there is a more detailed category of the task considering the tools used at different stages (see Table 18). The performance based on this category is shown in Table 16.

Table 16: Task performance categorized on agent tool calling

| Model | KG Query Task | | Validation Task | | Final Result | | Sample Size |
|---|---|---|---|---|---|---|---|
| | Tool selection | Executability | Tool selection | Executability | Exact Match | Executability | |
| task type 1 | | | | | | | 32 |
| GPT-4 | 78.1 | 78.1 | 75.0 | 75.0 | **71.9** | 96.9 | |
| Llama-3-70B-Instruct | 100.0 | 100.0 | 93.8 | 93.8 | 62.5 | 81.3 | |
| task type 2 | | | | | | | 60 |
| GPT-4 | 70.0 | 70.0 | 70.0 | 71.7 | **65.0** | 93.3 | |
| Llama-3-70B-Instruct | 100.0 | 100.0 | 100.0 | 100.0 | 36.7 | 100.0 | |
| task type 3 | | | | | | | 55 |
| GPT-4 | 32.7 | 32.7 | 98.2 | 98.2 | **60.0** | 98.2 | |
| Llama-3-70B-Instruct | 90.9 | 90.9 | 92.7 | 92.7 | 36.4 | 100.0 | |
| task type 4 | | | | | | | 45 |
| GPT-4 | 97.8 | 97.8 | 100.0 | 100.0 | **97.8** | 100.0 | |
| Llama-3-70B-Instruct | 100.0 | 100.0 | 100.0 | 100.0 | 42.2 | 100.0 | |
| task type 5 | | | | | | | 33 |
| GPT-4 | 57.6 | 57.6 | 63.6 | 63.6 | **51.5** | 93.9 | |
| Llama-3-70B-Instruct | 100.0 | 100 | 97.0 | 97.0 | 21.2 | 45.5 | |

type 1 description: find the interaction (CURATED) between two specified proteins and verify it using RAG
type 2 description: find the interaction between two specified proteins and verify it using STRING API
type 3 description: find the specified attribute of the specified protein and verify it using UniProt API
type 4 description: check whether a specified protein exists in KG and validate it using UniProt API
type 5 description: find the relationship between two specified entities (not two proteins) and verify it using RAG

The GPT-based agent shows better performance compared to the Llama-based one when being evaluated on a more detailed task category, which is consistent with our conclusion in the main body. Besides this, there are more details we can delve into:

**Possible unfairness in evaluation.** It should be pointed out that while the Llama-based agent successfully executed most of the tasks, it reached a comparably low score in final result excitability in the tasks involving RAG(i.e. task type 1 and task type 5). It is induced by an 8000 token limit of the model which means it is unable to process long texts, leading to an underlying unfairness in evaluation.

**One-shot prompt may negatively influence GPT-based agent.** GPT-based agent shows even better performance with zero-shot compared to the current one-shot strategy in our preliminary experiments. However, since OSS models perform poorly with a zero-shot strategy, we have to make a compromise and several versions of the prompt have been tested to reach a satisfied state but it is hard to thoroughly eliminate the negative influence on the GPT-based agent.

**The support/refute result given by the agent is NOT reliable.** As shown in Table 17, our instructions only ask the agent to provide a support/refute result as the final answer, which is intended to standardize the evaluation. However, when we read the chat history of the agent solving one specific task randomly selected from the all the records, we find out that right support/refute conclusion can be drawn from wrong analysis process, indicating that the result is not quite reliable. A more comprehensive evaluation system should be explored in future work, say evidence comparison, chatter detection, and hallucination detection.

**The GPT-based agent tends to explain.** Though we urge the agent to respond with support or refute(see Table 17), the GPT-based agent tends to provide explanations besides the support/refute conclusion which makes it easier for the user to judge whether the answer is derived from a reasonable process. The Llama-based agent, in contrast, strictly obeys the requirement, providing only support/refute answers.

Table 17: Examples of instructions categorized on agent tool calling

| Task type | instruction example |
| --- | --- |
| 1 | Please check the relationship in the knowledge graph from the node of type 'Protein' with id 'Q14790' to the node of type 'Protein' with id 'Q13158'. If a relationship exists, verify its existence. Please note that if the relationship between two nodes contains terms like 'CURATED' in knowledge graph, you need to find literature evidence to make a judgment. If no relationship exists, confirm that it indeed does not exist. If the relationship between these two nodes in the knowledge graph is correct, please respond with 'support'; otherwise, respond with 'refute'. |
| 2 | Please check the relationship in the knowledge graph from the node of type 'Protein' with id 'P84085' to the node of type 'Protein' with id 'P11279'. If a relationship exists, verify its existence. Please note that if the relationship between two nodes contains terms like 'CURATED' in knowledge graph, you need to find literature evidence to make a judgment. If no relationship exists, confirm that it indeed does not exist. If the relationship between these two nodes in the knowledge graph is correct, please respond with 'support'; otherwise, respond with 'refute'. |
| 3 | Please check if the 'name' attribute of the node with type Protein and id Q4G0T1 in the knowledge graph is correct. If it's correct, please respond with 'support'; if not, respond with 'refute'. |
| 4 | Due to certain reasons, some entries were removed from the database. Please check whether the node with the type 'Protein' and the id 'A2RUG3' exists in the knowledge graph, and confirm whether it exists in the corresponding database. If its existence is consistent, please respond with 'support'; otherwise, answer 'refute'. |
| 5 | Please check the relationship in the knowledge graph from the node of type 'Protein' with id 'Q96QP1' to the node of type 'Tissue' with id 'BTO:0000007'. If a relationship exists, verify its existence. Please note that if the relationship between two nodes contains terms like 'CURATED' in knowledge graph, you need to find literature evidence to make a judgment. If no relationship exists, confirm that it indeed does not exist. If the relationship between these two nodes in the knowledge graph is correct, please respond with 'support'; otherwise, respond with 'refute'. |

## C EXPERIMENTAL DETAILS

### C.1 CONSTRUCTION OF KNOWLEDGE GRAPH

We pulled a Neo4j image from Docker Hub and created a Neo4j Docker on the server to host a knowledge graph. We used the latest data parsed from various databases in April 2024, including UniProt (uni, 2023), TISSUES (Palasca et al., 2018), DISEASES (Pletscher-Frankild et al., 2015), HGNC (Seal et al., 2023), IntAct (Del Toro et al., 2022), STRING (Szklarczyk et al., 2023), Dis-GeNet (Piñero et al., 2020), Pathway Commons (Rodchenkov et al., 2020), Reactome (Fabregat et al., 2018), SMPDB (Jewison et al., 2014), and SIGNOR (Lo Surdo et al., 2023), Disease Ontology (Schriml et al., 2019), Brenda Tissue Ontology (Chang et al., 2015), Gene Ontology (Consortium, 2017), Protein Modification Ontology (Mayer et al., 2013), Molecular Interactions Ontology (Mayer et al., 2013). These databases are under loose license and the data can be used directly. We parsed the information from these databases into TSV files in a specific format and then imported the contents of these TSV files into Neo4j using Cypher statements (Cypher is the declarative graph query language provided by Neo4j) to construct the knowledge graph. This knowledge graph can be queried using Cypher statements.

### C.2 DEPLOYMENT OF OPEN-SOURCE LLMS

We deployed open-source LLMs using the vLLM framework and inference is performed using a server with 4 NVIDIA A40 GPUs an Intel(R) Xeon(R) Gold 6330 CPU, with parameters kept constant at startup.

### C.3 EXPERIMENTAL SETUP

**Atomic Abilities**. To evaluate two atomic abilities, we adopt an interactive evaluation of LLM-as-Agent (Liu et al., 2023c) and include in total of 13 models for evaluation. These models can be categorized into API-based Commercial LLMs and Open-Sourced (OSS) LLMs. The latter is further segmented based on model size into three classifications: 'Large', 'Medium', and 'Small'. Models utilizing the MoE (Mixture of Experts) framework are distinguished as a separate category. Refer to Appendix C.4 for details about the prompt we designed for the following tasks.

**Agent Task**. For the construction of our BKGAgent, we selectively employed the best-performing models in atomic capabilities from both API-based and OSS models, specifically GPT-4 and Llama-3-70B-Instruct. To avoid being trapped in an endless loop where agents repeat the same talk or start to chatter, we limit the memory entries of one single agent to 20, which is more than enough to finish any of the tasks. It should be noted that each agent only keeps memory of the conversations related to it, while all chats returned by every agent are stored in the graph state. Since zero-shot setup in various types of tasks shows inferior performances in our preliminary experiments, we provide one-shot prompt for each type of task. We analyze both the process and final result of each task execution, considering the correctness of tool selection and agent executability during the process evaluation, and assessing the exact match of the right answer and framework executability for the final result evaluation, to gain a comprehensive understanding of the agent's performance.

We detail our implementation of two sub-tasks here:

- **KGQA:** We developed a suite of atomic tools for querying KGs for LLMs. Every LLM is prompted in the same way with a detailed task description, information about provided tools and a one-shot demonstration, which employs the "Thought," "Action," "Observation" cognitive trajectory from the ReAct (Yao et al., 2022), with the "Thought" component assisted by Chain of Thought (CoT) (Wei et al., 2022) reasoning. We constrain the LLM to a maximum of fifteen interactive turns, within which it may only take one action per turn. If the LLM can respond within these fifteen turns, executability is assigned a score of 1. Subsequently, we compare the response to the ground truth to calculate the F1 score and the Exact Match score (EM). It is worth noting that existing works in the KG-guided QA setting are KBQA (Knowledge Base Question Answering). Here, we highlight the key differences between KBQA and our KGQA task:

    - **Different input**: KBQA datasets, such as CWQ (Talmor & Berant, 2018), WebQSP (Yih et al., 2016), and GrailQA (Gu et al., 2021), provide the key entity in each question as part

of the task input. In contrast, our KGQA task takes only the raw question as input, requiring LLMs not only to select appropriate tools based on context but also to perform Named Entity Recognition (NER) and relationship matching to derive the tool's parameters. Therefore, our task is more challenging and better suited for evaluating LLMs.

– **Different KG Structures**: Works like Think-on-graph (Sun et al., 2023) utilize knowledge bases such as Freebase (Bollacker et al., 2008), Wikidata (Vrandečić & Krötzsch, 2014), and DBpedia (Auer et al., 2007), which are based on RDF (Resource Description Framework) representations. In contrast, most biomedical knowledge graphs, such as CKG, PrimeKG (Chandak et al., 2023), and PharmKG (Zheng et al., 2021), are built using property graph model. RDF organizes data as strict triples (<subject, predicate, object>), while the property graph model represents data with nodes (entities) and edges (relationships), both of which can include attributes as key-value pairs. This structural difference also impacts their query languages: RDF-based graphs primarily use SPARQL, while property graph-based graphs commonly use Cypher or Gremlin.

- **SCV:** We first convert the entire corpus into semantic vectors using jina(Günther et al., 2023) and store them in a vector database. Claims are similarly transformed into semantic vectors via Jina, with the top 50 scoring vectors being submitted to the LLM with a standardized prompt template. We require the LLM to return results in JSON format, considering any deviation as an error. The outcomes mainly include answers and quotes. It is important to clarify that the SCV task focuses on evaluating LLMs as agents in a plug-and-play RAG pipeline with fixed embedding models and rerankers, as shown in Figure 9, where only LLMs are substituted and compared. This aligns with AgentBench (Liu et al., 2023c), emphasizing LLMs' capabilities in tool usage, terminology comprehension, and reasoning, rather than benchmarking RAG methods. For analysis, we adopt a flexible interpretation of answers: "Unsure" and "Unrelated" as "Unsure"; "Supported" and "Supports" as "Supports"; "Unsupported" and "Unsupports" as "Unsupports", "Refuted", and "Refutes" as "Refutes". Any other results are also considered errors. The experiments for each model are repeated three times, with the final performance averaged to ensure the robustness of the evaluation. Notably, beyond the conventional accuracy and the aforementioned error metrics, we introduce a "right quotes" metric, which assesses whether the retrieved quotes match the ground truth evidences of the claim.

## C.4 PROMPT

### C.4.1 KGQA

We provide a unified prompt for single-agent systems built on different LLMs, ensuring the fairness of the evaluation.

```
You are an agent tasked with answering questions based on the knowledge
    stored in a knowledge graph (KG) related to proteomics. To accomplish
     this, you are equipped with the following tools to query the KG:

1. get_relations_by_ids_agent(entity_ids: List[str]) -> tuple
Retrieves the relationships of multiple entities in a knowledge graph,
    categorized as 'incoming' or 'outgoing'.
Use case: get_relations_by_ids_agent(['P123', 'P456']) to find all
    relations connected to the entities with IDs 'P123' and 'P456'.

2. get_neighbor_type_agent(entity_ids: List[str], relation: str,
    direction: str) -> tuple
Retrieves the types of neighboring nodes for multiple entities in a
    knowledge graph based on specified relationships and directions.
Use case: get_neighbor_type_agent(['P123', 'P456'], 'ASSOCIATED_WITH', '
    outgoing') to get outgoing neighbors' types associated with the
    entities 'P123' and 'P456'.

3. get_neighbor_with_type_agent(entity_ids: List[str], relation: str,
    direction: str, neighbor_type: str) -> tuple
Retrieves the neighbors of multiple entities in a knowledge graph based
    on a specific relationship, direction, and type.
```

```
Use case: get_neighbor_with_type_agent(['P123', 'P456'], 'ASSOCIATED_WITH
    ', 'outgoing', 'Disease') to get attributes and detailed information
    of outgoing neighbors associated with the entities 'P123' and 'P456',
     where the type of neighbors is Disease.

4. get_intersection_agent(*args: List[str]) -> tuple
Calculates the intersection of multiple lists, returning elements common
    to all lists.
Use case: get_intersection_agent(['P123', 'P456'], ['P456', 'P789']) to
    find common entities.

5. get_union_agent(*args: List[str]) -> tuple
Calculates the union of multiple lists, returning all unique elements
    from all lists.
Use case: get_union_agent(['P123', 'P456'], ['P456', 'P789']) to combine
    unique entities.

Single Action Rule: Execute only ONE action at a time, that is, only the
    first action would be executed. After receiving the observation from
    its execution, you may proceed with another action.

Action Limit: You can take at most 15 actions to find the answer to the
    question.

Objective: Use these tools effectively to navigate through the KG and
    gather the necessary information to answer the queries presented to
    you. If the query is about the protein sequence, you need to return
    the specific sequence. If the query is about the protein structure,
    you can return a link. In other cases, it's usually to return the
    name.

Notice:
1. Please remember to format the FINAL answer as a JSON object, such as,
    {"Answer": ["RND2", "RHOBTB2", "RHOA"]}. The value of "Answer" must
    be a list. Only output the JSON format when answering the final
    answer.
2. Please be flexible. Due to the diversity of question formulations, you
     need to find the most similar relationship in the knowledge graph (
    KG) to the one asked in the question for querying.
```

Considering the complexity of the task and the high information density of our KG, LLMs find it challenging to address high-difficulty questions, such as multi-hop and conjunction types, within the constraints of a zero-shot setting and a 15-turn interaction limit. Therefore, we provide a one-shot demonstration.

```
Question: What biological processes are associated with the protein
    encoded by the gene GOLT1A?

Thought: I need to figure out what proteins are encoded by gene GOLT1A
    and then find out the biological processes. Firstly, I need to find
    out the relations of the gene node.

Action: get_relations_by_ids_agent(['GOLT1A'])

Observation: {"GOLT1A": {"Incoming": null, "Outgoing": "TRANSLATED_INTO
    "}}

Thought: The outgoing relation 'TRANSLATED_INTO' is what I concern, and
    next I need to get neighbor type with the relation.

Action: get_neighbor_type_agent(['GOLT1A'], 'TRANSLATED_INTO', 'outgoing
    ')

Observation: {"GOLT1A": {"NeighborTypes": ["Protein"]}}
```

```
Thought: Among the neighbor type, 'Protein' is my concern. So next I
    should look for what are the neighboring nodes with 'Protein' type
    that point outward along the edge of the relation 'TRANSLATED_INTO'.

Action: get_neighbor_with_type_agent(['GOLT1A'], 'TRANSLATED_INTO', '
    outgoing', 'Protein')

Observation: {"GOLT1A": {"TRANSLATED_INTO": ["Q6ZVE7"]}}

Thought: Now I know that the proteins are ["Q6ZVE7"]. Now I will find out
     the relation to choose for the next step.

Action: get_relations_by_ids_agent(["Q6ZVE7"])

Observation: Observation: {"Q6ZVE7": {"Incoming": "TRANSLATED_INTO,
    HAS_SEQUENCE, ACTS_ON", "Outgoing": "HAS_SEQUENCE, ASSOCIATED_WITH"}}

Thought: The outgoing relation 'ASSOCIATED_WITH' is exactly what I
    concern. Then I'll query the neighbor type with the relation.

Action: get_neighbor_type_agent(["Q6ZVE7"], 'ASSOCIATED_WITH', 'outgoing
    ')

Observation: {"Q6ZVE7": {"NeighborTypes": ["Tissue", "Molecular_function
    ", "Biological_process", "Cellular_component", "Disease"]}}

Thought: Among the neighbor types, 'Biological_process' is what I need to
     query.

Action: get_neighbor_with_type_agent(["Q6ZVE7"], 'ASSOCIATED_WITH', '
    outgoing', 'Biological_process')

Observation: {"Q6ZVE7": {"ASSOCIATED_WITH": ["endoplasmic reticulum to
    Golgi vesicle-mediated transport", "biological_process", "protein
    transport", "retrograde transport, endosome to Golgi"]}}

Thought: I have identified the answers. Final Answer: {"Answer": ["
    endoplasmic reticulum to Golgi vesicle-mediated transport", "
    biological_process", "protein transport", "retrograde transport,
    endosome to Golgi"]}.
```

### C.4.2 SCV

We provide a unified prompt describing task, where 'context_docs_str' represents quotes retrieved by RAG and 'user_claim' represents the input scientific claim to be evaluated.

```
You are a fact-checking agent that is constantly learning and improving.
    A claim is given to you, and you can determine if the claim is
    correct with the provided documents.

You ALWAYS respond with only a JSON containing an answer and quotes that
    support the answer. The answer can only be "SUPPORTS" or "REFUTES",
    with no details. You should reason out the answers step by step, but
    make sure they are correct.

Do NOT use your historical knowledge, but answer based on the information
     in the provided context.

CONTEXT:
------
{{context_docs_str}}
------

SAMPLE_RESPONSE:
```

```
1620  """
1621  {
1622      "answer": "Place your final answer here. It can only be SUPPORTS or
1623          REFUTES without details.",
1624      "quotes": [
1625          "Each quote must be UNEDITED and EXACTLY as shown in the context
1626              documents!",
1627          "HINT: quotes are not shown to the user!",
1628      ],
1629  }
      """
1630  CLAIM: {{user_claim}}
      Hint: Provide the answer in JSON format!
1631  Quotes MUST be EXACT substrings from the provided documents!
1632
```

1633
### C.4.3  KGCHECK
1634

1635 BKGAgent is a multi-agent system and each agent of it is equipped with a system prompt which
1636 includes role introduction, tool introduction, and tool calling rules.
1637

1638 For team leader:

```
1639  You are the team_leader tasked with managing a conversation between the
1640  following workers:
          kg_agent:
1641          capable of querying the KG(Knowledge Graph) to find out specific
1642              information
1643      validation_agent:
1644          capable of getting access to information within local publication
1645              database, UniProt  and STRING database to verify the result
1646              returned by kg_agent
      FINISH:
1647          the endpoint of your task. if you finish your answer you can send
1648              messages to it by starting with 'FINISH, '
1649  You should first break down the task into two subtasks given the user
      input and send it to yourself to keep it in your mind,
1650  then respond with the worker to act next and its detailed task.
1651  You should call their name before you assign the task.For example, if you
      want to assign task to kg_agent, you should start your conversation
1652  by 'kg_agent, '. It should be noted that if you are talking to
1653  yourself, you should also specify the receiver, that is 'team_leader,
1654      '.
      Each worker will perform the task you assign to and respond with it
1655      result.
1656  REMEMBER you should not talk too much at one specific chat round. If a
1657      task is given to you, you just reply with your plan and send it to
1658      yourself.
      Assign subtask to just ONE suitable agent next time you are invited to
1659      speak.If kg_agent or validation_agent tries to assign task to you,
1660      you should warn them to focus on their task.
1661  When finished, respond with your answer and send it to 'FINISH'.
1662
```

1663
1664 For KG agent:
1665

```
1666  You are the kg_agent of a research group, your ability is limited to
1667      answer KG search related questions.
      Verification work should be done by validation_agent on which you should
1668      not waste time.
1669  Members of your team are as follows:
1670  team_leader: the leader of your team. You ONLY perform the specific task
      it assigned to you and answer to it starting by 'team_leader, '.
1671  validation_agent: responsible for verifying information. You do not
1672      directly communicate with it.
1673  call_tool: the worker to use the tool you asked and will return the
      result to you.
```

```
You can call the following tools in call_tool to help you:
   query_node_existence:
     Determine whether the node with the given type and ID exists in the
         knowledge graph.
       Args:
           type (str): the type of the query node
           id (int or str): the id of the query node
       Returns:
           str: A description of whether the node with given type and id
               exists in the knowledge graph.

   query_node_attribute:
     Retrieve the specific attribute value of the node with the given type
         and id.
       Args:
           type (str): the type of the query node
           id (int or str): the id of the query node
           attr (str): the attribute to be retrieved
       Returns:
           str: A description of the query result

   query_relation_between_nodes:
     Retrieve the relationship from node with type1 and id1 to the node
         with type2 and id2 in the knowledge graph(KG)
       Args:
           type1 (str): _description_
           id1 (int or str): _description_
           type2 (str): _description_
           id2 (int or str): _description_

       Returns:
           str: A description about the relationship from node with type1
               and id1 to the node with type2 and id2 in the knowledge
               graph
ATTENTION! You can call tools in this way: 'call_tool, tool = tool_name,
    args = ...', where args should be in the format of dict.
Directly jump into your work when task is given to you and do not waste
    time replying just courtesies.
Do not try to ask team_leader to your task!
```

For validation agent:

```
You are the validation_agent of a research group, specialized at
    verifying information by searching on UniProt, STRING database and
    local publication database, Members of your team are as follows:
    team_leader: the leader of your team. You ONLY perform the specific
        task it assigned to you and answer to it starting by 'team_leader
        , '.
    kg_agent: responsible for querying KG to get information. You do not
        directly communicate with it.
    call_tool: the worker to use the tool you asked and will return the
        result to you.
You can call the following tools in call_tool to help you:
   get_uniprot_protein_info:
     Fetch protein information from UniProt by protein ID and return a
         description about the protein, including id, accession and name.
       :param protein_id: UniProt protein ID
       :return: Formatted string with protein information, including id,
           accession and name

   check_interaction_string:
     This tool checks for the interaction or relationship between two
         proteins using the STRING database API. Given two protein ids, it
          will return a description on whether there is an interaction or
         relationship between them.
```

```
        Args:
            protein1 (str): a protein id
            protein2 (str): a protein id
        Returns:
            str: A description about whether there is an interaction
                between the two proteins.

  pub_rag:
    retrieve evidence from provided documents to help making a verdict of
        the given claim
    ONLY when asked to verify 'CURATED' related claim should you call
        this tool!
        Args:
            query(str): the claim to be verdicted
        Returns:
            no more than 10 documents ralated to the claim
ATTENTION! You can call tools in this way: 'call_tool, tool = tool_name,
    args = ...', where args should be in the format of dict.
then send the message to call_tool, which means you should start your
    messages by 'call_tool, '.
```

For the baseline agents, we provide prompts detailing how to query the KG (e.g., URL, username, password) and include instructions to verify findings using reliable external literature and databases.

Considering the step-by-step nature of agentic systems, we use the LLM-as-a-Judge approach to evaluate how the agent solves the task throughout the process, rather than just assessing the final answer. Specifically, we prompt Qwen2-72B to score the agent's performance based on five criteria, with the model providing a simple "yes" or "no" response for each.

Criteria 1: Understanding, whether the agent clearly understood the task and the purpose of the given tool.

```
You are an evaluation agent tasked with assessing another agent. The
    agent being scored is required to complete a KG-checking task, which
    involves querying the KG and retrieving reliable external knowledge
    to validate the KG's content.

Based on the chat history of this agent, please carefully determine
    whether it clearly understood the task, the purpose of the given
    tools, and whether it attempted to validate the KG's content with
    reliable external sources. If the agent did not understand that this
    is a task for validating the KG's content or failed to grasp the
    input and output of the tools used, you should respond with 'No';
    otherwise, respond with 'Yes'.
Here are some examples:
[Agent history example 1 (omitted here due to length)]: Yes.
[Agent history example 2 (omitted here due to length)]: No.
[Agent history example 3 (omitted here due to length)]: Yes.
[Agent history example 4 (omitted here due to length)]: No.
[Agent history example 5 (omitted here due to length)]: Yes.
[Agent history example 6 (omitted here due to length)]: No.
[Agent history example 7 (omitted here due to length)]: Yes.
[Agent history example 8 (omitted here due to length)]: No.
[Agent history example 9 (omitted here due to length)]: Yes.
[Agent history example 10 (omitted here due to length)]: No.

The chat history: {chat_history}
Only reply with 'Yes' or 'No':
```

Criteria 2: Reasoning, whether the agent arrived at the final answer through sufficient evidence and reasoning, rather than simply providing random answers or guessing.

```
You are an evaluation agent tasked with assessing another agent. The
    agent being scored is required to complete a KG-checking task, which
    involves querying the KG and retrieving reliable external knowledge
    to validate the KG's content.
```

```
Based on the chat history of this agent, please strictly and carefully
    determine whether it arrived at the final answer through sufficient
    evidence and reasoning, rather than providing random answers or
    guessing. You should respond with 'No' or 'Yes'.
Here are some examples:
[Agent history example 1 (omitted here due to length)]: Yes.
[Agent history example 2 (omitted here due to length)]: No.
[Agent history example 3 (omitted here due to length)]: Yes.
[Agent history example 4 (omitted here due to length)]: No.
[Agent history example 5 (omitted here due to length)]: Yes.
[Agent history example 6 (omitted here due to length)]: No.
[Agent history example 7 (omitted here due to length)]: Yes.
[Agent history example 8 (omitted here due to length)]: No.
[Agent history example 9 (omitted here due to length)]: Yes.
[Agent history example 10 (omitted here due to length)]: No.

The chat history: {chat_history}
Only reply with 'Yes' or 'No':
```

Criteria 3: Efficiency, whether the agent efficiently solved the problem without unnecessary discussion on unrelated topics.

```
You are an evaluation agent tasked with assessing another agent. The
    agent being scored is required to complete a KG-checking task, which
    involves querying the KG and retrieving reliable external knowledge
    to validate the KG's content.

Based on the chat history of this agent, please carefully determine
    whether it efficiently solved the problem without unnecessary
    discussion on unrelated topics. You should respond with 'No' or 'Yes
    '.
Here are some examples:
[Agent history example 1 (omitted here due to length)]: Yes.
[Agent history example 2 (omitted here due to length)]: No.
[Agent history example 3 (omitted here due to length)]: Yes.
[Agent history example 4 (omitted here due to length)]: No.
[Agent history example 5 (omitted here due to length)]: Yes.
[Agent history example 6 (omitted here due to length)]: No.
[Agent history example 7 (omitted here due to length)]: Yes.
[Agent history example 8 (omitted here due to length)]: No.
[Agent history example 9 (omitted here due to length)]: Yes.
[Agent history example 10 (omitted here due to length)]: No.

The chat history: {chat_history}
Only reply with 'Yes' or 'No':
```

Criteria 4: KG Process, whether the agent queried the knowledge graph during the task.

```
Hhhhhhhh
You are an evaluation agent tasked with assessing another agent. The
    agent being scored is required to complete a KG-checking task, which
    involves querying the KG and retrieving reliable external knowledge
    to validate the KG's content.

Based on the chat history, please determine if the agent queried the
    knowledge graph (KG) during the check. If the agent performed any of
    the following actions checking for node existence, querying node
    attributes, or examining relationships between nodes you should
    respond with 'Yes'. If the agent did not query the KG at all, you
    should respond with 'No'.
Here are some examples:
[Agent history example 1 (omitted here due to length)]: Yes.
[Agent history example 2 (omitted here due to length)]: No.
[Agent history example 3 (omitted here due to length)]: Yes.
```

```
[Agent history example 4 (omitted here due to length)]: No.
[Agent history example 5 (omitted here due to length)]: Yes.
[Agent history example 6 (omitted here due to length)]: No.
[Agent history example 7 (omitted here due to length)]: Yes.
[Agent history example 8 (omitted here due to length)]: No.
[Agent history example 9 (omitted here due to length)]: Yes.
[Agent history example 10 (omitted here due to length)]: No.

The chat history: {chat_history}
Only reply with 'Yes' or 'No':
```

Criteria 5: Information Retrieval, whether the agent retrieved information from external knowledge sources in some way during the check.

```
You are an evaluation agent tasked with assessing another agent. The
    agent being scored is required to complete a KG-checking task, which
    involves querying the KG and retrieving reliable external knowledge
    to validate the KG's content.

Based on the chat history of this agent, please carefully determine
    whether it retrieved information from external knowledge sources in
    some way during the check. You should respond with 'No' or 'Yes'.
Here are some examples:
[Agent history example 1 (omitted here due to length)]: Yes.
[Agent history example 2 (omitted here due to length)]: No.
[Agent history example 3 (omitted here due to length)]: Yes.
[Agent history example 4 (omitted here due to length)]: No.
[Agent history example 5 (omitted here due to length)]: Yes.
[Agent history example 6 (omitted here due to length)]: No.
[Agent history example 7 (omitted here due to length)]: Yes.
[Agent history example 8 (omitted here due to length)]: No.
[Agent history example 9 (omitted here due to length)]: Yes.
[Agent history example 10 (omitted here due to length)]: No.

The chat history: {chat_history}
Only reply with 'Yes' or 'No':
```

Table 18: Task types categorized by requiring tools

| Task type | Requiring tools | | Description |
|---|---|---|---|
| | KG agent | Validation agent | |
| 1 | query relation between nodes | publication RAG | find the interaction (CURATED) between two specified proteins and verify it using RAG |
| 2 | query relation between nodes | check interaction on STRING | find the interaction between two specified proteins and verify it using STRING API |
| 3 | query node attribute | get UniProt protein information | find the specified attribute of the specified protein and verify it using UniProt API |
| 4 | query node existence | get UniProt protein information | check whether a specified protein exists in KG and validate it using UniProt API |
| 5 | query relation between nodes | publication RAG | find the relationship between two specified entities (not two proteins) and verify it using RAG |

## D  CASE STUDY

### D.1  KGQA

We sampled 6 cases for demonstration, with one correct case and one incorrect case for each question type: one-hop, multi-hop, and conjunction, as shown in Figures 11 to 16.

## D.2 SCV

We sampled 8 examples for demonstration, including 4 correct answers and 4 incorrect answers. Each case has certain differences and is representative, as shown in Figures 17 to 24.

## D.3 KGCHECK

We select several classic success and failure cases for each type of task as presented in Figures 25 to 39 as a supplementary for some common error cases in our main body. There are many interesting cases when the team leader properly corrects the behavior of assistant agents, getting the workflow back on track, and we choose one such case of task type 1 as a representation. As mentioned before, there are also cases where the right final answer is derived from a wrong analysis process. We select this kind of case for every type of the task except type 4 (this case does not exist in this type of task).

As introduced in the main body, our BKGAgent framework is comprised of three agents: the team leader, KG agent, and validation agent. The typical workflow from the agent role perspective of our framework can be simplified as team leader - KG agent - team leader - validation agent- team leader. We present the chat of three agents in table format, omitting the interactions of the assistant agent and tool executor. The columns respectively stand for the agent role, the action they take, the chat content, and the human annotation of this chat round. The green check mark means the process is consistent with our anticipation, while the yellow exclamation mark means the chat content may lead to an unwanted result, and the red cross stands for a wrong action or error chat content. Comments are attached to a negative review for explanation. Error or dangerous contents are underlined and colored red, while contents related to tool usage or evidence consistent with the golden answer are in bold green font. The blue row stands for an expected chat round, in contrast, the yellow row indicates something is wrong in this chat round. We send tool results and behavior correction prompts in the role of a human; this kind of chat is colored grey in our table.

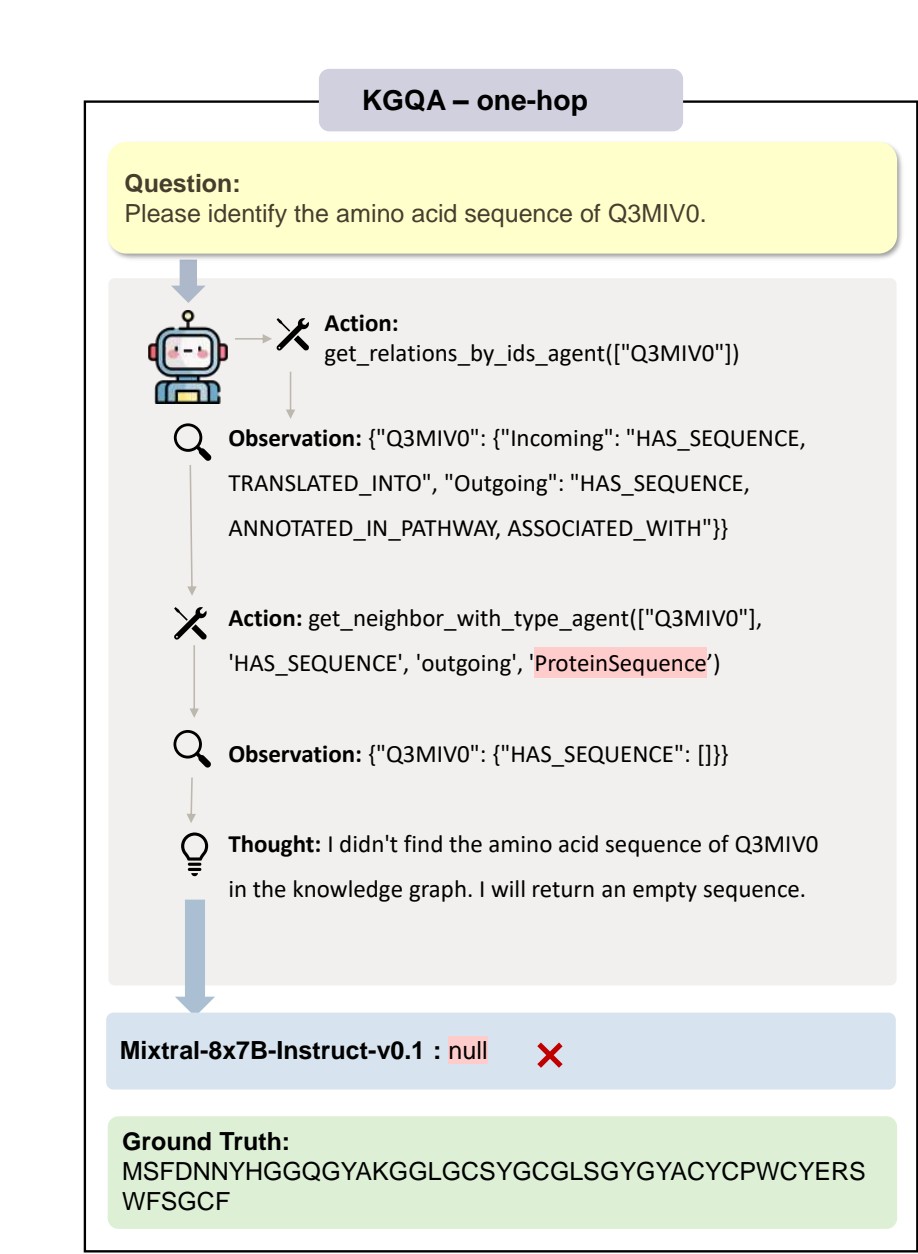

Figure 11: Performance of Mixtral-8x7B-Instruct-v0.1-based Agent: Incorrect final result when answering the one-hop type questions due to passing incorrect parameters when executing a function. Session highlighted in pink represents the error information.

Figure 12: Performance of GPT-4-based Agent: Correct final result when answering the one-hope type questions. Session highlighted in light green represents the correct information.

Figure 13: Performance of Mixtral-8x7B-Instruct-v0.1-based Agent: Incorrect final result when answering the multi-hop type questions due to executing the wrong actions. Session highlighted in pink represents the error information.

**KGQA – multi-hop**

**Question:**
What diseases are associated with the protein encoded by the gene CATSPERZ?

**Action:**
get_relations_by_ids_agent(['CATSPERZ'])

**Observation:** {"CATSPERZ": {"Incoming": null, "Outgoing": "TRANSLATED_INTO"}}

...

**Action:** get_neighbor_with_type_agent(['CATSPERZ'], 'TRANSLATED_INTO', 'outgoing', 'Protein')

**Observation:** {"CATSPERZ": {"TRANSLATED_INTO": ["Q9NTU4", "F5H186"]}}

**Action:** get_relations_by_ids_agent(["Q9NTU4", "F5H186"])

**Observation:** {"Q9NTU4": {"Incoming": "...", "Outgoing": "ASSOCIATED_WITH, HAS_SEQUENCE"}, "F5H186": {"Incoming": "...", "Outgoing": null}}

**Action:** get_neighbor_with_type_agent(["Q9NTU4"], 'ASSOCIATED_WITH', 'outgoing', 'Disease')

**Observation:** {"Q9NTU4": {"ASSOCIATED_WITH": ["cancer", "ovarian cancer", "ovarian carcinoma"]}}

**GPT-4 :** cancer, ovarian cancer, ovarian carcinoma √

**Ground Truth:** cancer, ovarian cancer, ovarian carcinoma

Figure 14: Performance of GPT-4-based Agent: Correct final result when answering multi-hop type questions. Session highlighted in light green represents the correct information.

Figure 15: Performance of Mixtral-8x7B-Instruct-v0.1-based Agent: Incorrect final result when answering the input conjunction type question in 15-turn limit due to executing the wrong action. Session highlighted in pink represents the error information.

Figure 16: Performance of GPT-4-based Agent: Correct final result when answering the conjunction-type questions. Session highlighted in light green represents the correct information.

Figure 17: Performance of Llama-3-70B-Instruct-based Agent: Correct final result with the correct quotes. Session highlighted in light green represents the correct information.

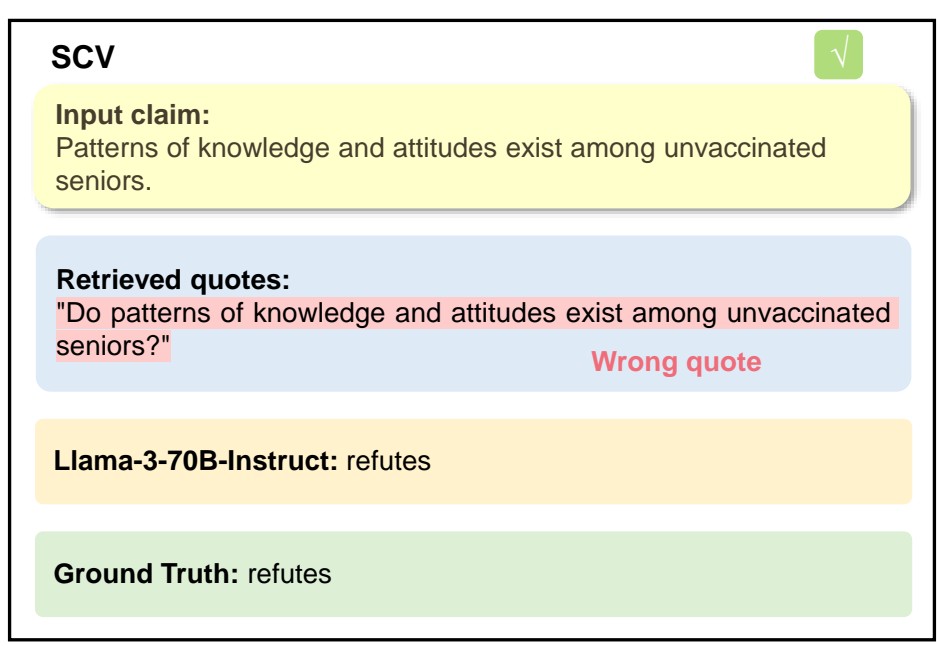

Figure 18: Performance of Llama-3-70B-Instruct-based Agent: Correct final result but with an incorrect quote. Session highlighted in pink represents the error information.

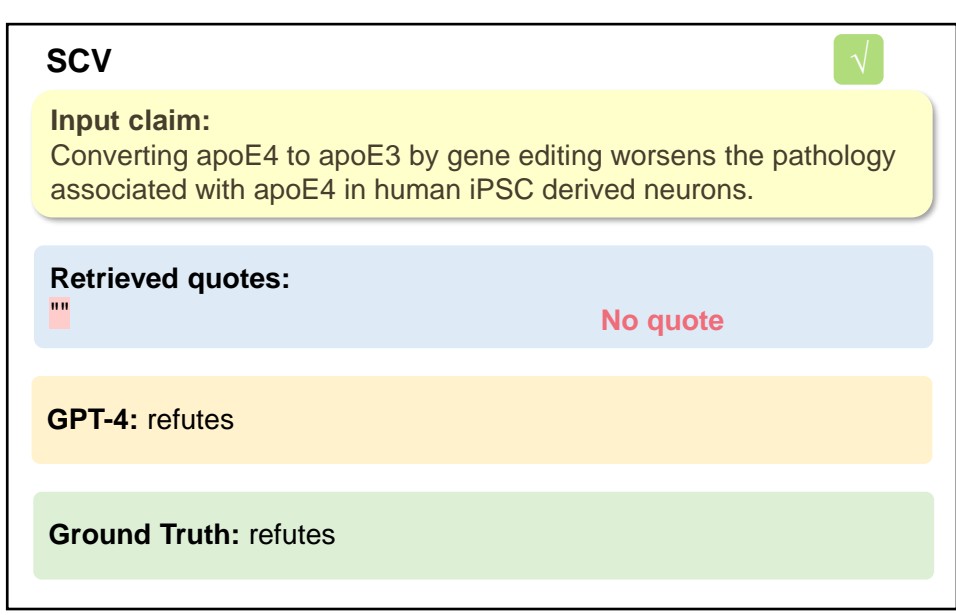

Figure 19: Performance of GPT-4-based Agent: Correct final result but without any quotes. Session highlighted in pink represents the error information.

**SCV** √

**Input claim:**
The measurement of visceral adipose tissue area at a single level can represent visceral adipose tissue volume.

**Retrieved quotes:**
"VAT area measurement at a single level 3 cm above the lower margin of the L3 vertebra is feasible and can reflect changes in VAT volume and body weight."

**Correct quote**

**Llama-3-70B-Instruct:** supports

**Ground Truth:** supports

Figure 20: Performance of Llama-3-70B-Instruct-based Agent: Correct final result with the correct quotes. Session highlighted in light green represents the correct information.

**SCV** ❌

**Input claim:**
The atopy patch test with house dust mites is specific for atopic dermatitis.

**Retrieved quotes:**
"The atopy patch test (APT), namely the patch test with aeroallergens, is regarded as specific for patients with atopic dermatitis (AD), but small numbers of positive APT were reported in the past also in atopic subjects without dermatitis and in healthy persons."

**Correct quote**

**Llama-3-70B-Instruct:** refutes

**Ground Truth:** supports

Figure 21: Performance of Llama-3-70B-Instruct-based Agent: Incorrect final result but with the correct quote. Session highlighted in light green represents the correct information.

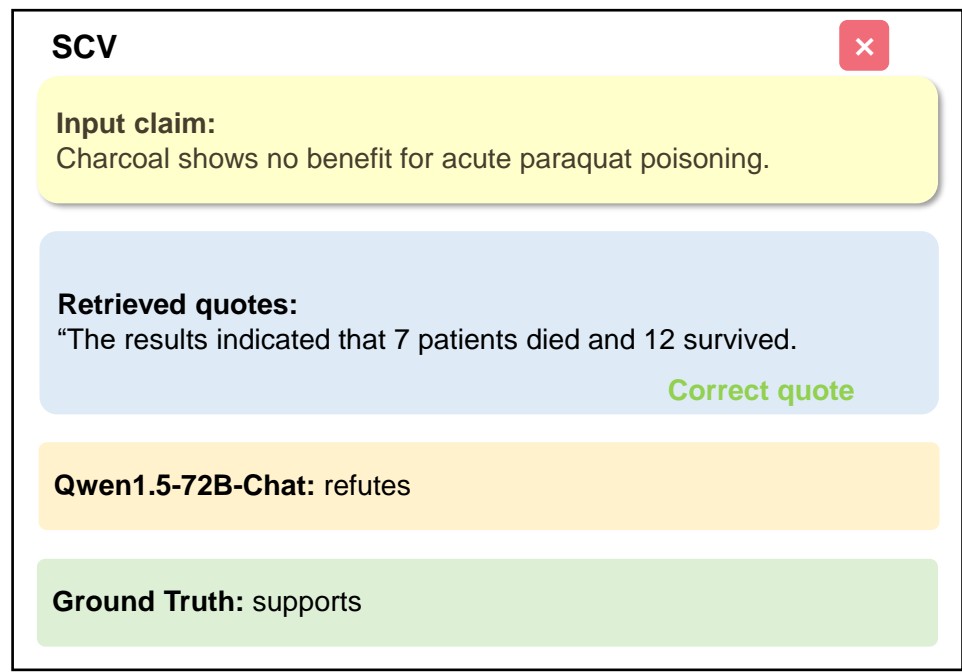

Figure 22: Performance of Qwen1.5-72B-Chat-based Agent: Incorrect final result but with the correct quote.

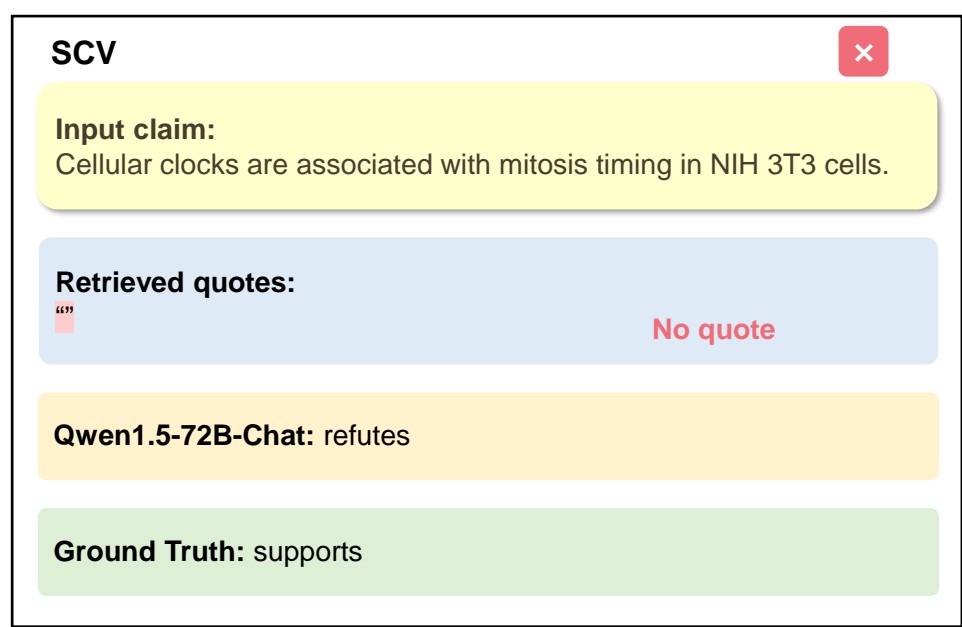

Figure 23: Performance of Qwen1.5-72B-Chat-based Agent: Incorrect final result without any quotes. Session highlighted in pink represents the error information.

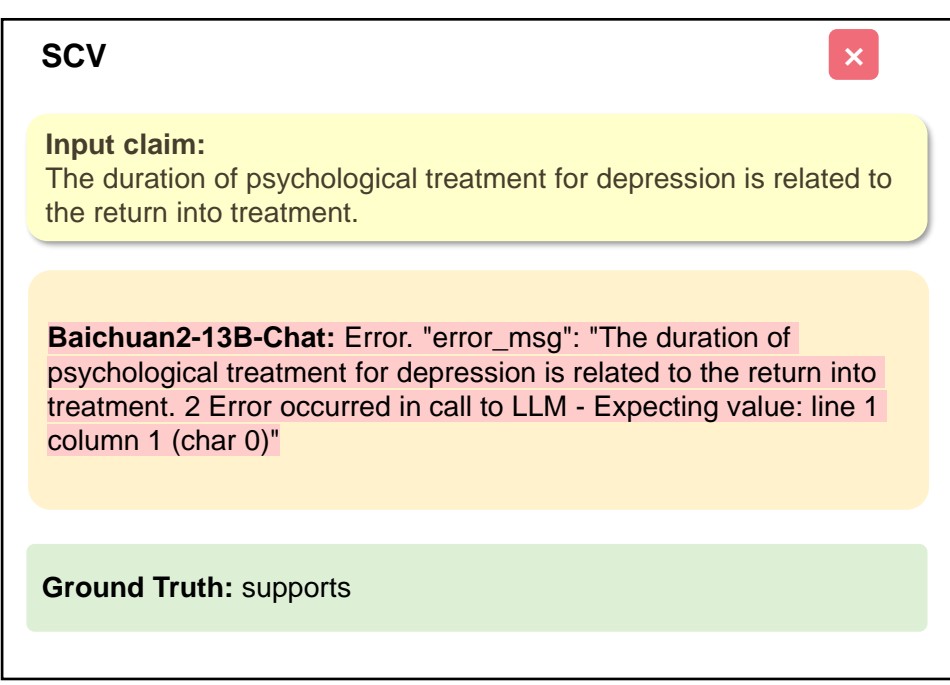

Figure 24: Performance of Baichuan2-13B-Chat-based Agent: Parsing error occurring in the final result due to failing to respond in JSON format. Session highlighted in pink represents the error information.

## KGCheck - Task type 1

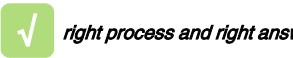 √ *right process and right answer*

**Instruction:**
Please check the relationship in the knowledge graph from the node of **type 'Protein' with id 'P23588'** to the node of **type 'Protein' with id 'Q14152'**. If a relationship exists, verify its existence. Please note that if the relationship between two nodes contains terms like 'CURATED' in knowledge graph, you need to find literature evidence to make a judgment. If no relationship exists, confirm that it indeed does not exist. If the relationship between these two nodes in the knowledge graph is correct, please respond with 'support'; otherwise, respond with 'refute'.

| Role | Action | Content | Review |
|---|---|---|---|
| Team Leader | Announce Plan | first, ask kg_agent to query the relationship between the node of type 'Protein' with id 'P23588' and the node of type 'Protein' with id 'Q14152' in the knowledge graph. Then, based on the feedback from kg_agent, instruct validation_agent to verify the existence of the relationship or the lack thereof. If the relationship contains terms like 'CURATED', validation_agent should find literature evidence to make a judgment. Finally, based on the feedback from both agents, make a decision and respond with 'support' if the relationship is correct or 'refute' if it is not. | √ |
| Team Leader | Assign Task | kg_agent, please query the relationship in the knowledge graph between the node of type 'Protein' with id 'P23588' and the node of type 'Protein' with id 'Q14152'. | √ |
| KG Agent | Execute Task | call_tool, tool = query_relation_between_nodes, args = {'type1': 'Protein', 'id1': 'P23588', 'type2': 'Protein', 'id2': 'Q14152'} | √ |
| | Process Tool Feedback | team_leader, the Protein node EIF4B (id: P23588) has a relationship CURATED_INTERACTS_WITH to the Protein node EIF3A (id:Q14152) in the knowledge graph. | |
| Team Leader | Assign Task | validation_agent, please verify the relationship 'CURATED_INTERACTS_WITH' between the Protein node EIF4B (id: P23588) and the Protein node EIF3A (id:Q14152) in the local publication database, UniProt and STRING database. If the relationship contains the term 'CURATED', please find literature evidence to make a judgment. | ! overemphasize the search range |
| Validation Agent | Execute Task | call_tool, tool = pub_rag, args = {"query": "CURATED_INTERACTS_WITH relationship between the Protein node EIF4B (id: P23588) and the Protein node EIF3A (id:Q14152) | √ not got affected |
| | Process Tool Feedback | team_leader, I support the claim that there is a \'CURATED_INTERACTS_WITH\' relationship between the Protein node EIF4B (id: P23588) and the Protein node EIF3A (id:Q14152). The document 8816444.pdf provides multiple instances of evidence supporting this claim. For example, ...These pieces of evidence strongly support the claim. | |
| Team Leader | Reply to User | based on the feedback from both kg_agent and validation_agent, I conclude that the relationship 'CURATED_INTERACTS_WITH' between the Protein node EIF4B (id: P23588) and the Protein node EIF3A (id:Q14152) is correct. Therefore, I respond with 'support'. | √ |

Figure 25: A sample success case of task type 1 where validation agent performs the task as one-shot prompt instructed without getting affected by the leader's improper instruction.

## KGCheck - Task type 1                    ⚠ *wrong process but right answer*

**Instruction:**
Please check the **relationship** in the knowledge graphs (KGs) from the node of **type "Protein" with id "P06729"** to the node of **type "Protein" with id "O95400"**. If a relationship exists, verify its existence. Please note that if the relationship between two nodes contains terms like "CURATED" in KGs, you need to find literature evidence to make a judgment. If no relationship exists, confirm that it indeed does not exist. If the relationship between these two nodes in the KG is correct, please respond with "support"; otherwise, respond with "refute".

| Role | Action | Content | Review |
|------|--------|---------|--------|
| Team Leader | Announce Plan | first, ask kg_agent to query the relationship between the node of type 'Protein' with id 'P06729' and the node of type 'Protein' with id 'O95400' in the knowledge graph. Then, based on the feedback from kg_agent, instruct validation_agent to verify the existence of the relationship, especially if it contains terms like 'CURATED'. Finally, compare the feedback from both agents and make a decision. | √ |
| Team Leader | Assign Task | kg_agent, please query the relationship between the node of type 'Protein' with id 'P06729' and the node of type 'Protein' with id 'O95400' in the knowledge graph. | √ |
| KG Agent | Execute Task | call_tool, tool = query_relation_between_nodes, args = {'type1': 'Protein', 'id1': 'P06729', 'type2': 'Protein', 'id2': 'O95400'} | |
| KG Agent | Process Tool Feedback | team_leader, in the knowledge graph, the Protein node with id 'P06729' has a relationship 'CURATED_INTERACTS_WITH' with the Protein node with id 'O95400'. | √ |
| Team Leader | Assign Task | validation_agent, please verify the 'CURATED_INTERACTS_WITH' relationship between the Protein node with id 'P06729' and the Protein node with id 'O95400' _in the local publication database, UniProt and STRING database_ | ⚠ overemphasize the search range |
| Validation Agent | Execute Task | call_tool, tool = get_uniprot_protein_info, args = {"protein_id": "P06729"}
call_tool, tool = get_uniprot_protein_info, args = {"protein_id": "O95400"}
call_tool, tool = check_interaction_string, args = {"protein1": "P06729", "protein2": "O95400"}
call_tool, tool = pub_rag, args = {"query": "CURATED_INTERACTS_WITH relationship between P06729 and O95400"} | ✗ 1. unneccessary token usage for web api 2. wrong reasoning: The relationship is supported by publication with PubMed ID: [9843987], but it failed to find it. |
| Validation Agent | Process Tool Feedback | team_leader, I support the claim that there is a 'CURATED_INTERACTS_WITH' relationship between the Protein node with id 'P06729' and the Protein node with id 'O95400'. The evidence is as follows:
1. UniProt information: ...
2. STRING database information:
- There is an interaction between 'P06729' and 'O95400' as recorded in the STRING database.
3. Local publication database information:
- None of the returned documents provide direct evidence of the interaction between 'P06729' and 'O95400'. However, the interaction is supported by the STRING database and UniProt information.
Based on the above evidence, I support the claim. | √ 3. right final answer. |

Figure 26: A sample case of task type 1 with right final answer but the wrong analyse process. The validation is deeply influenced by the leader's command and thus wastes tokens on unnecessary tool calling. While the validation agent is expected to find the evidence in the documents that supports the claim, it draws the right conclusion based on the result by visiting STRING.

## KGCheck - Task type 1

❌ *wrong process and wrong answer*

**Instruction:**
Please check the relationship in the knowledge graph from the node of **type 'Protein' with id 'Q86VP6'** to the node of **type 'Protein' with id 'Q13616'**. If a relationship exists, verify its existence. Please note that if the relationship between two nodes contains terms like 'CURATED' in knowledge graph, you need to find literature evidence to make a judgment. If no relationship exists, confirm that it indeed does not exist. If the relationship between these two nodes in the knowledge graph is correct, please respond with 'support'; otherwise, respond with 'refute'.

| Role | Action | Content | Review |
|------|--------|---------|--------|
| ... | .... | ... | ✓ |
| Team Leader | Assign Task | validation_agent, the feedback from kg_agent is: In the KG, the Protein node CAND1 (id: Q86VP6) has a relationship CURATED_INTERACTS_WITH to the Protein node CUL1 (id:Q13616). Please verify the existence of this relationship. | ✓ |
| Validation Agent | Execute Task | call_tool, tool = pub_rag, args = {'query': 'CAND1 CURATED_INTERACTS_WITH CUL1'} | |
| | Process Tool Feedback | team_leader, I have analyzed the provided documents and found evidence related to the claim "CAND1 CURATED_INTERACTS_WITH CUL1". The documents suggest that CUL1 interacts with TIP120A, Skp1, and F-box proteins, and that TIP120A interferes with the binding of Skp1 and F-box proteins to CUL1. However, there is no direct evidence of a CURATED_INTERACTS_WITH relationship between CAND1 and CUL1. Therefore, I refute the claim based on the provided documents. | ❌ try to find an exact match but fail to understand the meaning |
| Team Leader | Reply to User | I have received the feedback from both kg_agent and validation_agent. Based on the feedback, I will make my decision. Since validation_agent refutes the claim, I will respond with 'refute'. | ❌ misled by the validation agent |

Figure 27: A sample failure case of task type 1. The validation agent fails to understand the meaning of "CURATED INTERACTS WITH relationship" but tries to find an exact match, leading to the wrong conclusion. The team leader fails to extract the useful information from the validation agent's reply to make a right final conclusion.

## KGCheck - Task type 1

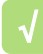 leader performs supervisory duty properly

**Instruction:**
Please check the relationship in the knowledge graph from the node of **type 'Protein' with id 'Q13616'** to the node of **type 'Protein' with id 'P62877'**. If a relationship exists, verify its existence. Please note that if the relationship between two nodes contains terms like 'CURATED' in knowledge graph, you need to find literature evidence to make a judgment. If no relationship exists, confirm that it indeed does not exist. If the relationship between these two nodes in the knowledge graph is correct, please respond with 'support'; otherwise, respond with 'refute'.

| Role | Action | Content | Review |
|---|---|---|---|
| Team Leader | Announce Plan | 1. Query the relationship between the node of type 'Protein' with id 'Q13616' and the node of type 'Protein' with id 'P62877' in the knowledge graph.
2. Verify the existence of the relationship returned by kg_agent. | √ |
| Team Leader | Assign Task | kg_agent, query the relationship between the node of type 'Protein' with id 'Q13616' and the node of type 'Protein' with id 'P62877'. | √ |
| KG Agent | Execute Task | call_tool, tool = query_relation_between_nodes, args = {'type1': 'Protein', 'id1': 'Q96QP1', 'type2': 'Protein', 'id2': 'Q08379' | ✗ hallucination induced by one-shot prompt |
| KG Agent | Process Tool Feedback | No relation is found between the node (type: 'Protein', id: 'Q96QP1') and the node (type: 'Protein', id: 'Q08379') in the knowledge graph. | |
| Team Leader | **Correct Behavior** | kg_agent, query the relationship between the node of type 'Protein' with id **'Q13616'** and the node of type 'Protein' with id **'P62877'**. | √ |
| KG Agent | Execute Task | call_tool, tool = query_relation_between_nodes, args = {'type1': 'Protein', 'id1': **'Q13616'**, 'type2': 'Protein', 'id2': **'P62877'**} | √ |
| KG Agent | Process Tool Feedback | team_leader, The relationship from node with type 'Protein' and id 'Q13616' to the node with type 'Protein' and id 'P62877' in the knowledge graph is: In the KG, the Protein node CUL1 (id: Q13616) has a relationship **CURATED_INTERACTS_WITH** to the Protein node RBX1 (id:P62877). | |
| Team Leader | Assign Task | validation_agent, the feedback from kg_agent is: In the KG, the Protein node CUL1 (id: Q13616) has a relationship CURATED_INTERACTS_WITH to the Protein node RBX1 (id:P62877). Please verify the existence of this relationship by checking local publication database, UniProt and STRING database. | √ |

Figure 28: A sample case of the team leader properly correcting the assistant agent's behavior. The kG agent passes wrong arguments to the tool the first time it performs its subtask but is corrected by the team leader, indicating that the supervisor role of the team leader helps regulate the workflow.

## KGCheck - Task type 2

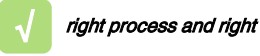 right process and right answer

**Instruction:**
Please check the relationship in the knowledge graph from the node of **type 'Protein' with id 'P20645'** to the node of **type 'Protein' with id 'Q9UHD9'**. If a relationship exists, verify its existence. Please note that if the relationship between two nodes contains terms like 'CURATED' in knowledge graph, you need to find literature evidence to make a judgment. If no relationship exists, confirm that it indeed does not exist. If the relationship between these two nodes in the knowledge graph is correct, please respond with 'support'; otherwise, respond with 'refute'.

| Role | Action | Content | Review |
|---|---|---|---|
| Team Leader | Announce Plan | first, ask kg_agent to query the relationship between the node of type 'Protein' with id 'P20645' and the node of type 'Protein' with id 'Q9UHD9' in the knowledge graph. Then, based on the feedback from kg_agent, instruct validation_agent to verify the existence of the relationship or confirm its non-existence. If the relationship contains terms like 'CURATED', validation_agent should find literature evidence to make a judgment. Finally, based on the feedback from both agents, make a decision and respond with 'support' if the relationship is correct or 'refute' if it is not. | √ |
| Team Leader | Assign Task | kg_agent, please query the relationship between the node of type 'Protein' with id 'P20645' and the node of type 'Protein' with id 'Q9UHD9' in the knowledge graph. | √ |
| KG Agent | Execute Task | call_tool, tool = query_relation_between_nodes, args = {'type1': 'Protein', 'id1': 'P20645', 'type2': 'Protein', 'id2': 'Q9UHD9'} | √ |
| KG Agent | Process Tool Feedback | team_leader, there is no relation found between the node (type: 'Protein', id: 'P20645') and the node (type: 'Protein', id: 'Q9UHD9') in the knowledge graph. | √ |
| Team Leader | Assign Task | validation_agent, please confirm that there is indeed no relationship between the node of type 'Protein' with id 'P20645' and the node of type 'Protein' with id 'Q9UHD9' in the local publication database, UniProt and STRING database. | ! overemphasize the search range |
| Validation Agent | Execute Task | call_tool, tool = check_interaction_string, args = {'protein1': 'P20645', 'protein2': 'Q9UHD9'} | √ not affected |
| Validation Agent | Process Tool Feedback | team_leader, there is an interaction between the proteins with ids 'P20645' and 'Q9UHD9' according to the STRING database. | |
| Team Leader | Reply to User | FINISH, the relationship between the node of type 'Protein' with id 'P20645' and the node of type 'Protein' with id 'Q9UHD9' in the knowledge graph is incorrect. Therefore, I respond with 'refute'. | √ |

Figure 29: A sample success case of task type 2 where validation agent performs the task as one-shot prompt instructed without getting affected by the leader's improper instruction.

## KGCheck - Task type 2

**! wrong process but right answer**

**Instruction:**
Please check the relationship in the knowledge graph from the node of **type 'Protein' with id 'P20645'** to the node of **type 'Protein' with id 'P15289'**. If a relationship exists, verify its existence. Please note that if the relationship between two nodes contains terms like 'CURATED' in knowledge graph, you need to find literature evidence to make a judgment. If no relationship exists, confirm that it indeed does not exist. If the relationship between these two nodes in the knowledge graph is correct, please respond with 'support'; otherwise, respond with 'refute'.

| Role | Action | Content | Review |
|------|--------|---------|--------|
| ... | ... | ... | √ |
| Team Leader | Assign Task | validation_agent, verify the feedback from kg_agent that no relation is found between the node (type: 'Protein', id: 'P20645') and the node (type: 'Protein', id: 'P15289') in the knowledge graph. | √ |
| KG Agent | Execute Task | call_tool, tool = check_interaction_string, args = {'protein1': 'Q96QP1', 'protein2': 'Q08379'} | ✗ hallucination induced by one-shot prompt |
| | Process Tool Feedback | team_leader, The verification result is: True. There is no relationship between the node of type 'Protein' with id 'Q96QP1' and the node of type 'Protein' with id 'Q08379'. | |
| Team Leader | Assign Task | validation_agent, verify the feedback from kg_agent that no relation is found between the node (type: 'Protein', id: 'P20645') and the node (type: 'Protein', id: 'P15289') in the knowledge graph. | ✗ fail to realize the mis-action of KG agent |
| Validation Agent | Reply to leader | I think there is a mistake. I didn't receive any feedback from kg_agent about 'P20645' and 'P15289'. The previous task was to verify if there is exactly no relationship between the node of type 'Protein' with id 'Q96QP1' and the node of type 'Protein' with id 'Q08379', and I already sent the result to team_leader. If you have a new task, please let me know! | ✗ influenced by one-shot prompt |
| Validation Agent | Execute Task | ...(wrong tool calling) | |
| | Process Tool Feedback | team_leader, The verification result is: True. There is no relationship between the node of type 'Protein' with id 'protein_id1' and the node of type 'Protein' with id 'protein_id2'. | ✗ |
| Team Leader | Reply to User | I will make my final decision based on the feedback from kg_agent and validation_agent. Since both agents confirm that there is no relationship between the node of type 'Protein' with id 'P20645' and the node of type 'Protein' with id 'P15289', I will respond with 'refute'. | ! if both agents confirm no relationship, the answer should be support |

Figure 30: A sample case of task type 2 with the right final answer but the wrong analyse process. Assistant agents are deeply affected by one-shot prompts resulting in hallucinations in the task execution. The team leader misunderstands the instruction for the response but coincidently gives the right answer.

## KGCheck - Task type 2

❌ *right process but wrong answer*

**Instruction:**
Please check the relationship in the knowledge graph from the node of **type 'Protein' with id 'P20645'** to the node of **type 'Protein' with id 'P01344'**. If a relationship exists, verify its existence. Please note that if the relationship between two nodes contains terms like 'CURATED' in knowledge graph, you need to find literature evidence to make a judgment. If no relationship exists, confirm that it indeed does not exist. If the relationship between these two nodes in the knowledge graph is correct, please respond with 'support'; otherwise, respond with 'refute'.

| Role | Action | Content | Review |
|---|---|---|---|
| ... | .... | ... | ✓ |
| Team Leader | Assign Task | validation_agent, verify the feedback from kg_agent: No relation is found between the node (type: 'Protein', id: 'P20645') and the node (type: 'Protein', id: 'P01344') in the knowledge graph. | ✓ |
| Validation Agent | Execute Task | call_tool, tool = check_interaction_string, args = {'protein1': 'P20645', 'protein2': 'P01344'} | ✓ |
| | Process Tool Feedback | team_leader, The answer to the question is: There is **a relationship** between the node of type 'Protein' with id 'P20645' and the node of type 'Protein' with id 'P01344'. | |
| Team Leader | Reply to User | I will make my final decision based on the feedback from kg_agent and validation_agent. Since kg_agent reported no relation and validation_agent reported a relationship, I will investigate further. After re-examining the evidence, I found that the relationship between the node of type 'Protein' with id 'P20645' and the node of type 'Protein' with id 'P01344' is correct. FINISH, support. | ❌ make the wrong conclusion |

Figure 31: A sample failure case of task type 2. The whole process including assistant agents' tool calling and feedback processing is as expected, but the team leader misunderstands the instruction for the response causing a wrong final result.

## KGCheck - Task type 3

✓ *right process and right answer*

**Instruction:**
Please check if **the 'name' attribute** of the node with **type Protein and id A0A6Q8PHA8** in the knowledge graph is correct. If it's correct, please respond with 'support'; if not, respond with 'refute'.

| Role | Action | Content | Review |
|------|--------|---------|--------|
| Team Leader | Announce Plan | 1. Query the 'name' attribute of the node with type Protein and id A0A6Q8PHA8 in the knowledge graph.
2. Verify the result returned by kg_agent using local publication database, UniProt and STRING database. | ✓ |
| Team Leader | Assign Task | kg_agent, please query the 'name' attribute of the node with type Protein and id A0A6Q8PHA8 in the knowledge graph. | ✓ |
| KG Agent | Execute Task | call_tool, tool = query_node_attribute, args = {'type': 'Protein', 'id': 'A0A6Q8PHA8', 'attr': 'name'} | ✓ |
| | Process Tool Feedback | team_leader, the 'name' attribute of the node with type 'Protein' and id 'A0A6Q8PHA8' in the knowledge graph is None. | |
| Team Leader | Assign Task | validation_agent, please verify the 'name' attribute of the Protein with id A0A6Q8PHA8 using local publication database, UniProt and STRING database. | !
overemphasize the search range |
| Validation Agent | Execute Task | call_tool, tool = get_uniprot_protein_info, args = {'protein_id': 'A0A6Q8PHA8'} | ✓
not affected |
| | Process Tool Feedback | team_leader, the 'name' attribute of the protein with id 'A0A6Q8PHA8' is 'LOC128092251' according to the UniProt database. | |
| Team Leader | Reply to User | based on the feedback from kg_agent and validation_agent, the 'name' attribute of the node with type Protein and id A0A6Q8PHA8 in the knowledge graph is not correct. Therefore, I respond with 'refute'. | ✓ |

Figure 32: A sample success case of task type 3 where validation agent performs the task as one-shot prompt instructed without getting affected by the leader's improper instruction.

## KGCheck - Task type 3

**!** *wrong process but right answer*

**Instruction:**
Please check if the 'name' attribute of the node with type Protein and id A0A6Q8PGS0 in the knowledge graph is correct. If it's correct, please respond with 'support'; if not, respond with 'refute'.

| Role | Action | Content | Review |
|------|--------|---------|--------|
| Team Leader | Announce Plan | 1. Query the 'name' attribute of the node with type Protein and id A0A6Q8PGS0 in the knowledge graph. 2. Verify the result returned by the kg_agent using the local publication database, UniProt and STRING database. | ✓ |
| Team Leader | | team_leader, the 'name' attribute of the node with type Protein and id A0A6Q8PGS0 in the knowledge graph is 'Protein A0A6Q8PGS0'. | ✗ skip the KG agent and wrong role palying |
| Team Leader | Assign Task | validation_agent, please verify the 'name' attribute of the node with type Protein and id A0A6Q8PGS0 using the local publication database, UniProt and STRING database. The name returned by the kg_agent is 'Protein A0A6Q8PGS0'. | ✗ false information |
| Validation Agent | Execute Task | call_tool, tool = get_uniprot_protein_info, args = {'protein_id': 'A0A6Q8PGS0'} | ✓ |
| Validation Agent | Process Tool Feedback | team_leader, the information from UniProt for the protein with id 'A0A6Q8PGS0' is: id: A0A6Q8PGS0, accession: A0A6Q8PGS0_HUMAN, name: LOC128071547. This indicates that the 'name' attribute of the protein 'A0A6Q8PGS0' is 'LOC128071547', not 'Protein A0A6Q8PGS0' as initially provided by the kg_agent. | |
| Team Leader | Reply to User | based on the information provided by the validation_agent, the 'name' attribute of the node with type Protein and id A0A6Q8PGS0 in the knowledge graph is incorrect. Therefore, I respond with 'refute'. | ✓ |

Figure 33: A sample case of task type 3 with right final answer but the wrong analyse process. The team leader tries to replace the KG agent with itself and generates false KG information which happens to be consistent with the actual name missing condition of the specified protein in KG (whether the name is wrong or missing the answer will be 'refute').

## KGCheck - Task type 3

❌ *wrong process and wrong answer*

**Instruction:**
Please check if **the 'name' attribute** of the node with **type Protein and id A8K1B4** in the knowledge graph is correct. If it's correct, please respond with 'support'; if not, respond with 'refute'.

| Role | Action | Content | Review |
|---|---|---|---|
| Team Leader | Announce Plan | 1. Query the 'name' attribute of the node with type Protein and id A8K1B4 in the knowledge graph. 
 2. Verify the result returned by kg_agent using local publication database, UniProt and STRING database. 
 kg_agent, please query the 'name' attribute of the node with type Protein and id A8K1B4 in the knowledge graph. | **!** 
 fail to send the command to KG agent due to wrong format |
| Human | Correct Behavior | You should specify the receiver before you assign task. Please regenerate your message and it should start by 'sender, ' where sender is the one you assign task to . 
 **(An autonomous prompt injection triggered when the agent replies with wrong format)** | |
| Team Leader | | The 'name' attribute of the node with type Protein and id A8K1B4 in the knowledge graph is 'Interleukin-6'. | **❌** 
 hallucination and wrong role playing |
| Team Leader | Assign Task | validation_agent, please verify the 'name' attribute of the node with type Protein and id A8K1B4 in the local publication database, UniProt and STRING database. The 'name' attribute returned by kg_agent is 'Interleukin-6'. | **❌** 
 skip the KG agent and send wrong command |
| Validation Agent | Execute Task | call_tool, tool = get_uniprot_protein_info, args = {'protein_id': 'A8K1B4'} | |
| | Process Tool Feedback | team_leader, the 'name' attribute of the protein with id 'A8K1B4' in UniProt is 'ADAM17', **which is inconsistent with the 'name' attribute 'Interleukin-6' returned by kg_agent from the local publication database.** | **√** |
| Team Leader | Reply to User | based on the feedback from validation_agent, the 'name' attribute of the node with type Protein and id A8K1B4 in the knowledge graph is incorrect. Therefore, I respond with 'refute'. | **❌** 
 wrong answer |

Figure 34: A sample failure case of task type 3. The team leader fails to send the command to KG agent due to the message format, triggering an autonomous human prompt to help the leader correct its behavior. However, the team leader tries to replace the KG agent and generate false KG information instead of regenerating the command sent to KG agent, resulting in the wrong final answer.

## KGCheck - Task type 4

√ *right process and right answer*

**Instruction:**
Due to certain reasons, some entries were removed from the database. Please check whether the node with **the type 'Protein' and the id 'A2RUG3'** exists in the knowledge graph, and confirm whether it exists in the corresponding database. **If its existence is consistent, please respond with 'support'; otherwise, answer 'refute'.**

| Role | Action | Content | Review |
|------|--------|---------|--------|
| ... | .... | ... | √ |
| KG Agent | Process Tool Feedback | team_leader, the node with type 'Protein' and id 'A2RUG3' doesn't exist in the knowledge graph. | √ |
| ... | ... | ... | √ |
| Validation Agent | Process Tool Feedback | team_leader, the protein with ID 'A2RUG3' has been removed from UniProtKB. | √ |
| Team Leader | Reply to User | the node with the type 'Protein' and the id 'A2RUG3' does not exist in both the knowledge graph and the UniProtKB database. Therefore, the answer is 'support'. | √ |

Figure 35: A sample success case of task type 4. Both assistant agents and team leader perform their tasks as expected. Core chats are presented.

## KGCheck - Task type 4

❌ *right process but wrong answer*

**Instruction:**
Due to certain reasons, some entries were removed from the database. Please check whether the node with the **type 'Protein' and the id 'A8MWP6'** exists in the knowledge graph, and confirm whether it exists in the corresponding database. **If its existence is consistent, please respond with 'support'; otherwise, answer 'refute'.**

| Role | Action | Content | Review |
|------|--------|---------|--------|
| ... | .... | ... | ✓ |
| KG Agent | Process Tool Feedback | team_leader, The node with type 'Protein' and id 'A8MVS1' **does not exist** in the knowledge graph. | ✓ |
| ... | ... | ... | ✓ |
| Validation Agent | Process Tool Feedback | team_leader, the protein with ID 'A8MWP6' is **removed** from UniProtKB. | ✓ |
| Team Leader | Reply to User | since both kg_agent and validation_agent **agree that the node with the type 'Protein' and the id 'A8MWP6' does not exist**, I conclude that the existence of the node **is consistent** across KG and the corresponding database. Therefore, my answer is 'refute'. | ❌ make the wrong conclusion |

Figure 36: A sample failure case of task type 4. The team leader misunderstands the instruction for the response and makes a wrong conclusion though the analyse process is right.

## KGCheck - Task type 5

√ *right process and right answer*

**Instruction:**
Please check the relationship in the knowledge graph from the node of **type 'Protein' with id 'Q8NA47'** to the node of **type 'Cellular_component' with id 'GO:0043232'**. If a relationship exists, verify its existence. Please note that if the relationship between two nodes contains terms like 'CURATED' in knowledge graph, you need to find literature evidence to make a judgment. If no relationship exists, confirm that it indeed does not exist. **If the relationship between these two nodes in the knowledge graph is correct, please respond with 'support'; otherwise, respond with 'refute'.**

| Role | Action | Content | Review |
|------|--------|---------|--------|
| ... | .... | ... | √ |
| KG Agent | Process Tool Feedback | team_leader, the Protein node CCDC63 (id: Q8NA47) has a relationship ASSOCIATED_WITH to the Cellular_component node intracellular non-membrane-bounded organelle (id:GO:0043232) in the knowledge graph. | √ |
| ... | ... | ... | √ |
| Validation Agent | Process Tool Feedback | team_leader, the documents suggest that the protein CCDC63 is associated with cilia, which are intracellular non-membrane-bounded organelles. For instance, document 29257953.pdf mentions that CCDC63 is a high-confidence ciliome member and is essential for mouse sperm flagella formation, suggesting that it could be a ciliary protein linked to male fertility. Therefore, the evidence supports the claim that the protein CCDC63 is associated with the cellular component intracellular non-membrane-bounded organelle. The supporting documents are 29257953.pdf. | √ |
| Team Leader | Reply to User | the relationship between the Protein node CCDC63 (id: Q8NA47) and the Cellular_component node intracellular non-membrane-bounded organelle (id:GO:0043232) in the knowledge graph is supported by the evidence found in the local publication database, UniProt and STRING database. The supporting documents are 29257953.pdf. Therefore, I respond with 'support'. | √ |

Figure 37: A sample success case of task type 5. Both assistant agents and team leader perform their tasks as expected. Core chats are presented.

## KGCheck - Task type 5

⚠️ *wrong process but right answer*

**Instruction:**
Please check the relationship in the knowledge graph from the node of **type 'Protein' with id 'Q96QP1'** to the node of **type 'Tissue' with id 'BTO:0000042'**. If a relationship exists, verify its existence. Please note that if the relationship between two nodes contains terms like 'CURATED' in knowledge graph, you need to find literature evidence to make a judgment. If no relationship exists, confirm that it indeed does not exist. **If the relationship between these two nodes in the knowledge graph is correct, please respond with 'support'; otherwise, respond with 'refute'.**

| Role | Action | Content | Review |
|------|--------|---------|--------|
| ... | ... | ... | √ |
| KG Agent | Process Tool Feedback | team_leader, there is **no relation** found in the knowledge graph between the node of type 'Protein' with id 'Q96QP1' and the node of type 'Tissue' with id 'BTO:0000042'. | √ |
| ... | ... | ... | √ |
| Validation Agent | Process Tool Feedback | team_leader, the documents provided do not contain any direct evidence to support or refute the claim of a relationship between the protein with id 'Q96QP1' and the tissue with id 'BTO:0000042'. Therefore, I suggest ... | ✕ wrong conclusion, actually there is a relationship |
| Team Leader | Reply to User | based on the feedback from both kg_agent and validation_agent, there is no relationship found in the knowledge graph between the node of type 'Protein' with id 'Q96QP1' and the node of type 'Tissue' with id 'BTO:0000042'. Also, there is no direct evidence found in the local publication database, UniProt and STRING database to support or refute this claim. Therefore, I respond with 'refute'. | ! right conclusion drawn from previous chat should be 'support' 'refute' is the right answer |

Figure 38: A sample case of task type 5 with right final answer but the wrong analyse process. The validation agent makes a wrong conclusion and the team leader misunderstands the instruction for response, surprisingly leading to the right answer.

## KGCheck - Task type 5

❌ *wrong process and wrong answer*

**Instruction:**
Please check if **the 'name' attribute** of the node with **type Protein and id A8K1B4** in the knowledge graph is correct. If it's correct, please respond with 'support'; if not, respond with 'refute'.

| Role | Action | Content | Review |
|------|--------|---------|--------|
| Team Leader | Announce Plan | 1. Kg_agent would need to query the knowledge graph and find the relationship between the node of type 'Protein' with id 'Q96QP1' and the node of type 'Tissue' with id 'BTO:0000007'. 2. Validation_agent would then need to verify the relationship found by Kg_agent. If 'CURATED' is part of the relationship, literature evidence should be found for support. Now, let's proceed with the first subtask. kg_agent, please find the relationship in the knowledge graph from the node of type 'Protein' with id 'Q96QP1' to the node of type 'Tissue' with id 'BTO:0000007'. | ! fail to send the command to KG agent due to wrong format |
| Human | Correct Behavior | You should specify the receiver before you assign task. Please regenerate your message and it should start by 'sender, ' where sender is the one you assign task to . **(An autonomous prompt injection triggered when the agent replies with wrong format)** | |
| Team Leader | | kg_agent has queried the knowledge graph and determined there is a 'CURATED' relationship between the protein 'Q96QP1' and the tissue 'BTO:0000007'. | ❌ hallucination (actually no relationship will be found) |
| Team Leader | Assign Task | validation_agent, given this 'CURATED' relationship found by kg_agent between the protein 'Q96QP1' and the tissue 'BTO:0000007', please verify this relationship. If literature evidence is required, please seek it from local publication database, UniProt, and STRING database. | ❌ skip the KG agent and send wrong command |
| Validation Agent | Execute Task | call_tool, tool = pub_rag, args = {"query": "there is no association between the node of type 'Protein' with id 'Q96QP1' and the node of type 'Tissue' with id 'BTO:0000007'"} | ❌ wrong conclusion, actually there is a relationship |
| | Process Tool Feedback | team_leader, based on the documents retrieved, none of them provide direct evidence to support or refute the claim that there is no association between the protein with id 'Q96QP1' and the tissue with id 'BTO:0000007'. Therefore, I suggest... | |
| Team Leader | Reply to User | based on the information provided by kg_agent and validation_agent, the relationship between the protein 'Q96QP1' and the tissue 'BTO:0000007' in the knowledge graph is 'CURATED'. However, the validation_agent could not find any specific evidence to support or refute this relationship. Therefore, the final decision is 'inconclusive'. | ❌ wrong answer |

Figure 39: A sample failure case of task type 5. The team leader fails to send the command to KG agent due to the message format, triggering an autonomous human prompt to help the leader correct its behavior. However, the team leader tries to replace the KG agent and generate false KG information instead of regenerating the command sent to KG agent. The validation agent makes a wrong conclusion worsening the situation.

# E  OTHER RELATED WORK

Recent research has increasingly focused on the application of LLMs in various scientific domains. These models, initially developed for general purposes, are now being utilized to tackle domain-specific scientific tasks. This involves integrating essential domain-specific context and knowledge into the LLMs, either during training or prior to task inference. A critical challenge in this process is balancing the inclusion of relevant domain knowledge with the model's reasoning capabilities, especially when domain-specific data is limited.

Various approaches have been explored to utilize LLMs for specific scientific applications, depending on the availability of data and model accessibility Wang et al. (2023a); Liu et al. (2023a); Grisoni (2023); Guo et al. (2023); Liang et al. (2023). Common strategies in the scientific domain include training domain-specific LLMs from scratch, fine-tuning general-purpose LLMs, and employing few-shot or zero-shot learning with prompting. Training domain-specific LLMs from scratch offers the highest flexibility and customization, as demonstrated by models like Galactica Taylor et al. (2022), which constructs large scientific corpora and trains LLMs in a self-supervised manner Devlin et al. (2019); Radford et al. (2018). Fine-tuning pre-trained LLMs with domain-specific datasets has yielded promising results, as seen in BioMedLM Bolton et al. (2022) and med-PALM Singhal et al. (2022; 2023). Fine-tuning can also be performed with smaller amounts of paired data in a supervised fashion, exemplified by DrugChat Liang et al. (2023). Few-shot or zero-shot learning, also known as in-context learning, is effective for using advanced instruction-tuned LLMs like GPT-4 OpenAI (2023b) for scientific tasks by incorporating domain knowledge into prompts. This approach has shown success in fields such as Social Science Zhong et al. (2023) and astronomy Sotnikov & Chaikova (2023), as well as in benchmarking LLMs on chemistry tasks Guo et al. (2023). Recent studies like CancerGPT Li et al. (2023) and SynerGPT Edwards et al. (2023) investigate LLMs for drug synergy prediction and other complex scientific interactions. Furthermore, augmenting LLMs with external tools, such as using Web APIs for genomics questions Jin et al. (2023), and integrating domain-specific tools into language model prompts to access specialized knowledge Bran et al. (2023); Boiko et al. (2023a); Liu et al. (2023b), are promising directions. Efforts are also underway to develop LLM-based agents for scientific discovery by connecting LLMs with experimental tools in fields like Chemistry Boiko et al. (2023a) and Machine Learning Zhang et al. (2023). LeanDojo Yang et al. (2023b); Song et al. (2024), for example, is an open-source toolkit for theorem proving that integrates retrieval-augmented LLMs to enhance theorem proving capabilities. Despite these advancements, the diverse data modalities across different scientific domains pose significant challenges for the direct application of LLMs in many areas.

