# OpenReview forum: "BioKGBench: A Knowledge Graph Checking Benchmark of AI Agent for Biomedical Science"
_ICLR.cc/2025/Conference — Submitted to ICLR 2025_

### Official Review · Reviewer_dk1Q · 2024-10-26

**Soundness:** 3
**Presentation:** 3
**Contribution:** 3
**Rating:** 6
**Confidence:** 3

**Summary:**

This work investigates the challenge of "ai agent for scientific literature" that focus solely on factual QA, leading to LLM hallucinations. A novel benchmark BioKGBench which integrates a Knowledge Graph Question Answering task and a Scientific Claim Verification task is proposed. Detailed construction (data source, statistics etc.) and a baseline is proposed to validate the quality of BioKGBench.

**Strengths:**

a) A novel benchmark called BioKGBench for evaluating biomedical ai agent is proposed.

b) Extensive evluation on existing ai agents are compared.

c) A baseline called BKGAgent is proposed.

**Weaknesses:**

a) Lack cost analysis of BKGAgent.

b) The generalizability of proposed pipeline to other domains is not discussed.

c) Minor writing issues.

**Questions:**

a) How the labels are made for SCV task? Any human agreement tests on the labeling?

b) Do the authors have a vision on the difference between proposed agent method and the existing agent methods? How the difference can impact on the performance on BioKGBench?

c) Do the authors have a vision on how the generalizability of proposed pipeline to other domains?

d) In line 363-364, the authors may want to avoid using the word "propose" since the metrics used are common.

e) How the classification is made for SCV task? Is it a self-critique / prompting process or a trainable process? If it's a prompt-based process, what is the scoring criteria?

---

> ### Author Response · Authors · 2024-11-22
> **Response to Reviewer dk1Q**
>
> Dear reviewer dk1Q,
>
> Thank you for your comments. I would like to address your concerns point by point.
>
> > **Weakness 1: Lack cost analysis of BKGAgent.**
>
> Thank you for your suggestion. We have conducted a cost analysis based on agent histories, calculating the average tokens per task, as shown in the table below. AutoGPT, despite being a single-agent system, has higher costs than AutoGen (dual-agent) due to its use of the **ReAct** algorithm. Our BKGAgent incurs **higher costs** due to its **three-agent** framework and the need for detailed **tool usage instructions** in prompts. This aligns with our discussion in the manuscript (lines 509–522).
>
> | Agent  | AutoGen | AutoGPT | BKGAgent (ours) |
> |---------|---------|---------|---------|
> | Average Tokens per task | 2225.7 | 2892.4 | 4006.3 |
>
>
> > **Weakness 2 & Question 3: The generalizability of proposed pipeline to other domains**
>
> Our work specifically **focuses on the biomedical domain** rather than building a general agent. When designing BKGAgent, we carefully considered the unique characteristics of biomedical knowledge and data, enabling it to process **KGs**, **databases**, and **research papers** effectively. Our pipeline is **generalizable to most biomedical KGs**, as they typically adopt a property graph model. Additionally, the generalizability of BKGAgent can be further enhanced by **extending its specialized tools** to accommodate other domain-specific requirements.
>
>
> > **Weakness 3 & Question 4: Minor writing issues with the use of the word "propose".**
>
> Thank you for your suggestion. We will replace the word "propose" with "adopt" in the revised PDF.
>
>
> > **Question 1: How the labels are made for SCV task? Any human agreement tests on the labeling?**
>
> The SCV dataset is reconstructed from **well-known, expert-annotated datasets** (lines 249-251). Additionally, we performed a **secondary verification using Qwen1.5-72B** to ensure the claims are consistent and free from conflicts.
>
>
> > **Question 2: Do the authors have a vision on the difference between proposed agent method and the existing agent methods? How the difference can impact on the performance on BioKGBench?**
>
> Yes, we discuss the differences and their impact primarily in Section 4.2:
> - **Atomic capabilities**: In Table 6, we compare the atomic abilities of BKGAgent with existing agents, highlighting that many existing agents cannot query KGs or access external knowledge bases, which makes them inadequate for tasks like KGCheck (lines 445–447).
> - **General abilities and specialized tools**: In the Agent Comparison section (lines 458–470), we emphasize that existing agents lack powerful tools compared to BKGAgent, resulting in poor performance. This highlights the importance of integrating general capabilities, such as coding, with specialized tools to improve agent effectiveness.
> - **Numerical differences**: In lines 509–522, we analyze the impact of increasing the number of agents, concluding that while it slightly improves performance, it also raises communication costs and complexities, which can offset the benefits.
>
>
> > **Question 5: Classification process and scoring criteria for SCV task.**
>
> As mentioned in lines 246-247, LLMs perform scientific claim verification using a **Retrieval-Augmented Generation (RAG)** approach. As detailed in Figure 3 and Appendix C.3, we fix the RAG pipeline with state-of-the-art embedding models and rerankers, feeding retrieval results into the prompts of LLMs, which then classify the claims as "support," "refute," or "nei." We evaluate LLM outputs using **Accuracy**, **Right Quotes**, and **Error Rate** (see Table 4, lines 363-367).
>
> We hope our response has adequately addressed your concerns. We appreciate the time and effort you have invested in reviewing our work.
>
> Best regards,
>
> Paper 7569 Authors

---

### Official Review · Reviewer_X1bo · 2024-11-03

**Soundness:** 2
**Presentation:** 3
**Contribution:** 3
**Rating:** 5
**Confidence:** 4

**Summary:**

This paper introduces BioKGBench, which explores the LLMs’ capability to do “KGCheck” - checking existing knowledge graph and cross-referencing with external literature or databases. They divide this task into two atomic sub-tasks: (1) knowledge graph question answering (KGQA), where the agent needs to answer questions of a knowledge graph; and (2) scientific claim verification (SCV), where the agent verifies a claim by querying the evidence from external datasets or literature.

To solve the tasks, this paper proposes BKGAent, which is a multi-agent framework containing a team leader, KG agent and validation agent equipped with a set of tools. The proposed framework achieves better performance than existing baselines.

**Strengths:**

1.	The sub-task design of KGCheck aligns well with the methodology of human scientific research, which consists of database queries and literature review. This benchmark may be useful for future study.
2.	The proposed BKGAgent incorporates various tools and interactions between agents, and achieves better performance than baselines.

**Weaknesses:**

1.	From Figure 5 and Table 8, the performance of “BKGAgent” is similar to that of “AutoGen w/ our tools”. It seems that the key performance gain is achieved by the tools. Thus, the effectiveness of using three agents in this framework is not clear.
2.	The first sub-task – KGQA is closely related to the body of work on KBQA. However, the relation with KBQA is not discussed in this paper. Also, is it possible to benchmark KBQA methods in the KGQA subtask and use them instead of KG agents in the KGCheck task?
3.	The performance of proposed framework on the sub-tasks are not reported and compared with baselines (Table 7).
4.	The metrics (understanding, reasoning, efficiency, KG process, information retrieval) are not explained and it is not clear what these scores imply.

**Questions:**

1.	What is the proportion of different reasoning types (one-hop, multi-hop, conjunction) involved in the KGCheck task?
2.	What is the recall rate of errors detected by BKGAgent on the KGCheck task?

---

> ### Author Response · Authors · 2024-11-21
> **Response to Reviewer X1bo (1/2)**
>
> Dear reviewer X1bo,
>
> Thank you for your comments. I would like to address your concerns point by point.
>
> > **Weakness 2: Discuss the relationship between KGQA and KBQA and the feasibility of using KBQA methods in the KGQA and KGCheck tasks.**
>
> Here, we highlight the key differences between KBQA and our KGQA task:
> - **Different Input**:
> KBQA datasets, such as CWQ, WebQSP, and GrailQA, **provide the key entity** in each question as part of the task input. In contrast, our KGQA task takes **only the raw question** as input, requiring LLMs not only to select appropriate tools based on context but also to perform **Named Entity Recognition (NER)** and relationship matching to derive the tool's parameters. Therefore, **our task is more challenging and better suited for evaluating LLMs**.
>
> - **Different KG Structures**: Works like Think-on-graph utilize knowledge bases such as Freebase, Wikidata, and DBpedia, which are based on **RDF (Resource Description Framework)** representations. In contrast, the vast majority of knowledge graphs in the biomedical domain, such as CKG, are built using **property graph model**. These two types of knowledge graphs differ significantly in terms of data storage models and query languages:
>     - **Data Models**: RDF enforces a strict **triple-based format** <subject, predicate, object> for representing data, whereas **property graph model** stores data in the form of nodes (entities) and edges (relationships), where **both nodes and edges can have attributes stored as key-value pairs**. This fundamental difference leads to distinct querying strategies.
>     - **Query Languages**: RDF-based knowledge graphs rely on **SPARQL** as the query language, whereas the latter uses **Cypher**.
>
> **These differences make it infeasible to directly benchmark KBQA methods like Think-on-graph on our KGQA task.**
>
>
> > **Weakness 4: Explaination of the metrics for the process.**
>
> We apologize for the lack of detailed explanation of the process metrics. In the revised PDF, we will include the following clarifications and add the prompts used in the appendix:
>
> - **Understanding**: whether the agent clearly understood the task and the purpose of the given tool.
> - **Reasoning**: whether the agent arrived at the final answer through sufficient evidence and reasoning, rather than simply providing random answers or guessing.
> - **Efficiency**: whether the agent efficiently solved the problem without unnecessary discussion on unrelated topics.
> - **KG Process**: whether the agent queried the knowledge graph during the task.
> - **Information Retrieval**: whether the agent retrieved information from external knowledge sources in some way during the check.
>
> As mentioned in lines 452-457, we employ Qwen2-72B to score the agent's performance across these five criteria, where the model provides a simple "yes" or "no" response for each criterion.
>
> > **Question 1: The proportion of different reasoning types (one-hop, multi-hop, conjunction) involved in the KGCheck task.**
>
> The KGCheck task focuses on **atomic checks**, which are classified into **node-level** and **triple-level** checks (as detailed in Table 5 and Lines 286–297). Since **one-hop, multi-hop, and conjunction reasoning ultimately reduce to triple-level checks**, we do not categorize them separately. For triple-level checks, we classify them into two types:
> - **Triple-existing**: Ensures the accuracy of relationships between connected nodes (25.8%).
> - **Triple-potential**: Assesses whether a relationship exists between unconnected nodes (29.8%).

---

> ### Author Response · Authors · 2024-11-23
> **Response to Reviewer X1bo (2/2)**
>
> > **Question 2: The recall rate of errors detected by BKGAgent on the KGCheck task.**
>
> As shown in the table below, our BKGAgent achieves a **20% higher recall rate of errors** (i.e., the ability to correctly identify errors in the KG) than the second-best agent, demonstrating its strong performance.
>
> | Agent  | Vanilla AutoGPT | AutoGPT w/ our tools | Vanilla AutoGen | AutoGen w/ our tools | BKGAgent (ours) |
> |---------|---------|---------|---------|---------|---------|
> | Recall Rate of Errors (%) | 50.0 | 45.6 | 1.1 | 61.1 | **81.1** |
>
> > **Weakness 1: The effectiveness of using three agents.**
>
> While our BKGAgent shows only a slight performance improvement over “AutoGen w/ our tools” in terms of EM, it achieves a **20% higher recall rate of errors** (i.e., the ability to correctly identify errors in the KG) than the second-best system, demonstrating its strong performance.
>
> Additionally, regarding the slight EM improvement, as a benchmark study, we also explored **how varying the number of agents impacts performance** to derive meaningful insights. As discussed in lines 509–522:
> - Increasing the number of agents raises **communication costs**, limiting significant performance gains.
> - Effective **planning** algorithms have a more substantial impact than the number of agents, as shown in Figure 5 (**6.3%** vs. 0.9%).

---

> > ### Comment · Reviewer_X1bo · 2024-11-26
> >
> > Thank you for the responses. They have addressed some of my concerns. However, I believe the current rating reflects the merits and shortcomings appropriately and I would keep it so far.

---

### Official Review · Reviewer_pqUW · 2024-11-04

**Soundness:** 2
**Presentation:** 2
**Contribution:** 2
**Rating:** 5
**Confidence:** 4

**Summary:**

This paper introduces BioKGBench, a benchmark for evaluating AI agents in biomedical science, designed to tackle the challenges of understanding and verifying scientific knowledge. Unlike traditional benchmarks that focus on narrow tasks like question-answering, BioKGBench emphasizes two critical abilities: Knowledge Graph Question Answering (KGQA) and Scientific Claim Verification (SCV). These tasks help assess agents' abilities to analyze both structured (knowledge graphs) and unstructured (scientific literature) data.
The authors also introduce BKGAgent, a baseline model that effectively identifies inaccuracies in biomedical knowledge graphs, offering a useful tool for keeping scientific databases accurate and up-to-date.

**Strengths:**

1. The paper proposes three tasks—Knowledge Graph Question Answering (KGQA), Scientific Claim Verification (SCV), and Knowledge Graph Checking (KGCheck).

2. The study successfully identifies 96 inaccuracies within the knowledge graph, marked as "Refute" annotations. This finding is particularly significant because it challenges the common assumption that knowledge graphs contain precise information.

**Weaknesses:**

1. The proposed method utilizes the Retrieval-Augmented Generation (RAG) approach; however, it lacks comparison with recent RAG-based baselines such as Self-RAG and KG-RAG. This absence makes it challenging to evaluate the effectiveness of the proposed method against state-of-the-art techniques, potentially limiting the credibility of the results.

2. The KGCheck task is primarily focused on verifying knowledge within knowledge graphs. While this is valuable, its narrow application may restrict the overall utility of the task in broader contexts.

3. The approach may face scalability issues when applied to larger and more complex knowledge graphs.

4. The study relies solely on one KG proposed in 2020, which may not reflect the most current state of knowledge representation. Given that the KG used is relatively old, there is a risk that the findings regarding inaccuracies and errors may not accurately reflect the current capabilities and improvements in more recent KGs. As KG evolve, their accuracy and completeness can change, potentially rendering the results of this study less relevant over time.

5. The results of the study may not be easily generalized to other KGs beyond the one used. Different KGs can vary significantly in structure, content, and accuracy, making it uncertain whether the identified issues and the effectiveness of the proposed tasks will hold true across various knowledge representations.

**Questions:**

1. What's the difference between KGQA and STaRK [1]?

2. Clinical Knowledge Graph used in the paper has difference references including [32], [33], and [53] in the paper.

[1] Wu, Shirley, Shiyu Zhao, Michihiro Yasunaga, Kexin Huang, Kaidi Cao, Qian Huang, Vassilis N. Ioannidis, Karthik Subbian, James Zou, and Jure Leskovec. "STaRK: Benchmarking LLM Retrieval on Textual and Relational Knowledge Bases." arXiv preprint arXiv:2404.13207 (2024).

---

> ### Author Response · Authors · 2024-11-21
> **Response to Reviewer pqUW (1/2)**
>
> Dear reviewer pqUW,
>
> Thank you for your comments. I would like to address your concerns point by point.
>
> > **Weakness 2: Limited applicability of the benchmark.**
>
> The KGCheck task combines two atomic tasks—KGQA and SCV. LLM agents are required to **identify errors in large-scale KGs** by processing both structured data (such as knowledge graphs) and unstructured data (such as literature). These are exactly the types of data typically generated by biomedical research. KGCheck evaluates the agent’s ability to comprehend and handle biomedical knowledge from heterogeneous sources, which we believe is highly valuable in this context and **offers insights into how LLM agents can be used to detect and correct errors in large-scale datasets**.
>
>
> > **Weakness 3&5: Scalability concerns for larger knowledge graphs and limited generalizability to other KGs.**
>
> Our approach is **scalable** in biomedical domain.
> Our approach is specifically designed for **property graph models**, which are **widely adopted in the construction of biomedical KGs**, such as BioKG, PharmKG, PrimeKG, etc. This characteristic makes our approach inherently scalable to other property graph-based knowledge graphs in the biomedical domain.
> We chose the CKG because it is one of the most authoritative and comprehensive biomedical KGs (line 197-206). As shown in following table, **CKG's scale and complexity are larger than many other well-known KGs**. This ensures that our experiments are conducted on a representative and sufficiently complex graph.
> Given that **property graph models dominate the biomedical domain** and that **we have validated our approach on a large and complex graph like CKG**, we believe our method is scalable and well-suited for biomedical knowledge graphs.
>
> Table. Comparison of different biomedical knowledge graphs.
>
> | KG Dataset  | Entity Types | Relation Types | Constituent Datasets | Journal | Property Graph Model |
> |---------|---------|---------|---------|---------|---------|
> | BioKG [1]   | 10 | 17 | 13 | ACM CIKM | √ |
> | PharmKG [2] | 3 | 29 | 7 | Briefings in Bioinformatics | √ |
> | PrimeKG [3] | 10 | 29 | 20 | Nature Scientific Data | √ |
> | **CKG [4] (ours)** | **35** | **57** | **35**  | Nature Biotechnology | √ |
>
>
>
> > **Weakness 4: Concerns over using an outdated knowledge graph.**
>
> As noted in **lines 298–300**, the CKG used in our study is built using database data from **May 2024, not 2020**. To ensure consistent benchmarking, we fix the KG data's timeframe, and the dataset labels are updated annually with the latest database releases to maintain relevance and accuracy.
>
>
>
> > **Question 2: Difference references of CKG.**
>
> Thank you for pointing this out, and we apologize for this minor writing issue. We will address it in the revised PDF and ensure the references are corrected.
>
>
>
>
> [1] Walsh, Brian, Sameh K. Mohamed, and Vít Nováček. "Biokg: A knowledge graph for relational learning on biological data." Proceedings of the 29th ACM International Conference on Information & Knowledge Management. 2020.
>
> [2] Zheng, Shuangjia, et al. "PharmKG: a dedicated knowledge graph benchmark for bomedical data mining." Briefings in bioinformatics 22.4 (2021): bbaa344.
>
> [3] Chandak, Payal, Kexin Huang, and Marinka Zitnik. "Building a knowledge graph to enable precision medicine." Scientific Data 10.1 (2023): 67.
>
> [4] Santos, Alberto, et al. "A knowledge graph to interpret clinical proteomics data." Nature biotechnology 40.5 (2022): 692-702.

---

> ### Author Response · Authors · 2024-11-23
> **Response to Reviewer pqUW (2/2)**
>
> > **Weakness 1: Lack comparison with other RAG-based baselines.**
>
> It is important to clarify that the SCV task is not designed to evaluate different RAG methods but rather to **evaluate LLMs as agents** within a plug-and-play RAG pipeline, using **fixed SOTA embedding models and rerankers**. As shown in Figure 9:
> - Step 1: Input claims and the corpus are embedded using a fixed embedding model and stored in a vector database.
> - Step 2: A reranker retrieves the top-k relevant bibliographies.
> - Step 3: The claim and retrieved bibliographies are fed to the LLM via a prompt template, and the LLM generates conclusions with logical quotations. In this step, we **substitute and compare different LLMs to evaluate their performance.**
>
> In conclusion, **RAG serves as a tool for LLMs** in our setup, not the primary subject of evaluation. Our study aligns with **AgentBench** [1] setup, aiming to assess LLMs' abilities in the biomedical domain, including **tool usage**, **terminology comprehension**, **reasoning**, rather than benchmarking RAG implementations.
>
>
> > **Question 1: Difference between KGQA and STaRK.**
> - **Setup**: **STaRK evaluates RAG methods** using various embedding models and rerankers, some of which are outdated now. In contrast, **RAG serves as a tool for LLMs in our setup**. We fix the RAG pipeline with state-of-the-art components (Appendix C.3, Figure 9) and use it to feed retrieval results into the prompts of LLMs, allowing the LLM to perform claim verification. Our primary focus is to **evaluate the LLM agent's capabilities, such as tool usage, biomedical knowledge understanding, and reasoning**.
> - **Tasks**: STaRK emphasizes retrieval, while our work goes further by requiring LLMs to **comprehend biomedical knowledge and reasoning to perform claim verification and make discoveries**.
>
> We hope our response has adequately addressed your concerns. We appreciate the time and effort you have invested in reviewing our work.
>
> [1] Liu, Xiao, et al. "Agentbench: Evaluating llms as agents." arXiv preprint arXiv:2308.03688 (2023).

---

### Official Review · Reviewer_1J1A · 2024-11-06

**Soundness:** 1
**Presentation:** 3
**Contribution:** 2
**Rating:** 3
**Confidence:** 4

**Summary:**

This paper proposes several tasks related to the scientific literature review and evaluates the performance of LLMs and LLM-based agents. The authors also propose an agent to address the task. These tasks include KG QA, scientific claim verification and the combined KG checking tasks. Lots of mainstream LLMs are tested. Several agent designs are also evaluated with access to the tools defined by the authors.

**Strengths:**

* The constructed test cases for KG QA and claim verification might be helpful in evaluating clinical-related models and agents
* The designed evaluation mechanism would be used as a good reference for future agent evaluation design

**Weaknesses:**

- Novelty as a benchmark work:
	- When evaluating the LLMs only (Table 7), the LLMs do not have access to the KG or literature database, making the task a straightforward task. Similar tasks/datasets have been proposed a lot
	- When evaluating the agents (Table 8), the assumption is that the tool design is not part of the agent design (as the same set of tools are provided by the authors instead of designed by the designer of the baseline agents). This assumption makes this benchmark not generalizable and limited by the design of tools used to access KG and literature
	- The agent benchmark task is not designed to be result-oriented
- Comparison with works of the KG-guided QA setting, such as Think-on-graph [1]
	- Both lines of work assume that external knowledge sources can be used to answer questions. What is the uniqueness of this work besides applying KG-enhanced QA to the biomedical domain?
- Soundness of the evaluation metrics
	- Why the potential relationship is refute if the relationship is not on the KG (which could totally due to the sparsity of the KG)? What is the difference between existing relationship and potential relationship?
- Missing details
	- For the claim verification task, when predicting label for a claim, does the model have access to the while corpus with 5664 articles, or it only have access to the particular article that the claim is induced from?
	- If this task is about global claims (instead of context-specific claims per article), how to handle the potential conflicts between claims?
	- What is the additional information the Qwen model used for evaluation has access to (compared with the evaluated LLMs that produce the answers)?
	- Does the framework support unseen nodes on the KG, or does it assume all entities used for evaluation have to be on the KG already?
	- For the baseline agent without access to the "our tools", how do they access the KG and literature pool?

[1] https://arxiv.org/abs/2307.07697

**Questions:**

Please refer to the questions in the "Weaknesses" section

---

> ### Author Response · Authors · 2024-11-21
> **Response to Reviewer 1J1A (1/3)**
>
> Dear reviewer 1J1A,
>
> Thank you for your comments. I would like to address your concerns point by point.
> - **Weakness 1: Novelty as a benchmark work.**
>     > When evaluating the LLMs only (Table 7), the LLMs do not have access to the KG or literature database, making the task a straightforward task.
>
>     As stated in the paper (lines 234-235 and 246-247), **the LLMs do have access to the knowledge graph (KG) and literature database during the evaluation.** This follows the **interactive evaluation of LLM-as-Agent** setup from AgentBench [1]. Specifically, for KGQA, we provide each LLM with the same tools via identical prompts. For SCV, we develop a plug-and-play retrieval-augmented generation (RAG) pipeline for the LLMs. These two tasks systematically evaluate an LLM's abilities in the biomedical domain, including tool usage, terminology understanding, reasoning, and more.
>
>     > When evaluating the agents (Table 8), the assumption is that the tool design is not part of the agent design.
>
>     We would like to clarify that we did not assume tools are separate from the agent system; rather, **tools are a critical part of the system**. In Table 8, we evaluate both the existing agent with our tools and the vanilla agent without our tools. Our results show that our **tools enhance performance**, while also demonstrating that a general agent, relying solely on coding abilities, cannot effectively handle specialized tasks in professional domains.
>
>     > The agent benchmark task is not designed to be result-oriented
>
>     Each task in our benchmark has been designed with specific **result-oriented evaluation metrics**, as outlined in **Table 1** of the paper. These metrics were carefully selected to quantitatively assess the performance of the agents based on the results they produce, ensuring a clear measure of success for each task.
>
> [1] Liu, Xiao, et al. "Agentbench: Evaluating llms as agents." arXiv preprint arXiv:2308.03688 (2023).

---

> ### Author Response · Authors · 2024-11-21
> **Response to Reviewer 1J1A (2/3)**
>
> - **Weakness 2: Comparison with works of the KG-guided QA setting.**
>
>   Existing works in the KG-guided QA setting are **KBQA** (Knowledge Base Question Answering). Here, we highlight the key differences between KBQA and our KGQA task:
>     - **Different Input**: KBQA datasets, such as CWQ, WebQSP, and GrailQA, **provide the key entity** in each question as part of the task input. In contrast, our KGQA task takes **only the raw question** as input, requiring LLMs not only to select appropriate tools based on context but also to perform **Named Entity Recognition (NER)** and relationship matching to derive the tool's parameters. Therefore, **our task is more challenging and better suited for evaluating LLMs**.
>     - **Different KG Structures**: Works like Think-on-graph utilize knowledge bases such as Freebase, Wikidata, and DBpedia, which are based on **RDF (Resource Description Framework)** representations. In contrast, the vast majority of knowledge graphs in the biomedical domain, such as CKG, are built using **property graph model**. These two types of knowledge graphs differ significantly in terms of data storage models and query languages:
>         - **Data Models**: RDF enforces a strict **triple-based format** <subject, predicate, object> for representing data, whereas **property graph model** stores data in the form of nodes (entities) and edges (relationships), where **both nodes and edges can have attributes stored as key-value pairs**. This fundamental difference leads to distinct querying strategies.
>         - **Query Languages**: RDF-based knowledge graphs rely on **SPARQL** as the query language, whereas the latter uses **Cypher**.
>
>   **These differences make it infeasible to directly benchmark KBQA methods like Think-on-graph on our KGQA task.**
>
> - **Weakness 3: Soundness of the evaluation metrics.**
>     - To clarify (also see lines 293-297 and Table 5), an **existing relationship** refers to a relationship that is already represented by an edge between two nodes in the KG. In this case, we check **whether the known relationship between the two entities is correct**. A **potential relationship**, on the other hand, refers to the absence of an edge between two nodes in the KG, which suggests that no relationship exists between the entities. In this case, we examine **whether a relationship actually exists between these two nodes**.
>     - In our task, an agent verifies KG triples using reliable external sources (**as ground truth**). If the agent finds a **contradiction** between the KG and external literature or databases, we label this as "refute". For example, if an agent identifies a relationship between two unconnected entities based on reliable papers, it indicates an error in the KG, which we define as "refute".

---

> ### Author Response · Authors · 2024-11-21
> **Response to Reviewer 1J1A (3/3)**
>
> - **Weakness 4: Missing details.**
>     > For SCV, does the model have access to the whole corpus with 5664 articles, or it only have access to the particular articles?
>
>     Thank you for your comment. To clarify, the LLMs have access to the **whole corpus** (the abstracts of 5,664 articles) when performing the claim verification task. We will add this clarification in the revised PDF.
>
>     > If this task is about global claims, how to handle the potential conflicts between claims?
>
>     As shown in Table 4, the claims are derived from the experimental results in the papers. The SCV dataset is reconstructed from **well-known, expert-annotated datasets** (lines 249-251). Additionally, we performed a **secondary verification using Qwen1.5-72B** to ensure the claims are consistent and free from conflicts. Therefore, we do not expect any conflicts between the claims.
>
>     > What is the additional information the Qwen model used for evaluation has access to?
>
>     As mentioned in **lines 454-456** of the paper, we provide the Qwen2-72B model with a prompt that includes several scoring demonstrations, the histories of the agents to be evaluated, and the evaluation criteria (five aspects).
>
>     > Does the framework support unseen nodes on the KG?
>
>     Our BKGAgent does support unseen nodes on the KG. However, our task only requires the agents to handle existing nodes in the KG.
>
>     > For the baseline agent without access to the "our tools", how do they access the KG and literature pool?
>
>     For the baseline agents, we provide prompts detailing how to query the KG (e.g., URL, username, password) and include instructions to verify findings using reliable external literature and databases.
>
> We hope our response has adequately addressed your concerns. We appreciate the time and effort you have invested in reviewing our work.

---

### Author Response · Authors · 2024-11-27
**General Response**

We sincerely appreciate all the reviewers for their time and detailed reviews. We have addressed the points of confusion raised by the reviewers, conducted additional experiments, and made appropriate modifications to the paper. Here is the summary of the highlights of our work, additional results, and revisions.

**Summary of Contributions and Novelty**
- **The First Benchmark for Evaluating Agents on Biomedical Knowledge Grounding**: We introduce BioKGBench with three tasks spanning structured and unstructured biomedical data, driving agents toward specialized scientific domains. (Reviewer dk1Q)
- **Novel KGCheck Task and Discovery Scenarios**：We identified KG errors with 96 “Refute” annotations, offering valuable data for agents to integrate heterogeneous knowledge and make discoveries. (Reviewer pqUW and X1bo)
- **Our Proposed BKGAgent Performs SOTA**: Our BKGAgent outperforms mainstream agents, supported by extensive experiments and analyses. (Reviewer X1bo and dk1Q)

**Summary of Additional Results**
- **Cost Analysis of Agents**: Compared the costs of different agents on the KGCheck task, yielding results consistent with the original analysis.
- **Evaluation of Agents' Recall Rate in KGCheck:**: Added an evaluation of agents' recall rate of errors in KGCheck, demonstrating BKGAgent's strong performance.

**Summary of Revisions:**
- **[Reviewer 1J1A and X1bo]**: Added a comparison between KBQA and our KGQA (Appendix C.3).
- **[Reviewer 1J1A and X1bo]**: Added an explanation of the process metrics adopted in KGCheck (understanding, reasoning, efficiency, KG process, information retrieval) and their corresponding prompts for the LLM evaluator (lines 449-455, Appendix C.4.3).
- **[Reviewer 1J1A]**: Added details about some experimental setup and data (lines 363, 1206-1207, Appendix C.3).
- **[Reviewer X1bo]**: Replace Figure 5 with a more informative version, incorporating the recall rate of errors metric and providing additional analysis (page 10).
- **[Reviewer pqUW and dk1Q]**: Corrected inappropriate terms and citation errors (lines 113, 199, 360-361).

All revisions are highlighted in blue.

---

### Meta-Review · Area_Chair_7c7b · 2024-12-20

**Metareview:**

This paper introduces BioKGBench, a novel benchmark designed to evaluate the capability of LLMs and AI-driven agents in biomedical science. The study distinguishes itself by attempting to address the challenges of factual verification and literature grounding in a systematic way.

However, as pointed out by the reviewers, there are several concerns with the methodology and generalizability of the findings: the simplicity of some tasks, which LLMs can already perform with high accuracy, questions the usefulness of such benchmarks; the use of a single, potentially outdated knowledge graph may not accurately represent current knowledge states, thus limiting the applicability of the results; and the assumption that tools provided by the authors, rather than developed by agent designers, limits the benchmark's generalizability. Even though the authors tried to address these concerns during the rebuttal, the reviewers were not fully satisfied with the responses.

**Additional Comments On Reviewer Discussion:**

Nil

---

### Decision · Program_Chairs · 2025-01-22

Reject